# How do LLMs Compute Verbal Confidence?

**Dharshan Kumaran** [1]   **Arthur Conmy** [1]   **Federico Barbero** [1]   **Simon Osindero** [1]   **Viorica Patraucean** [1]
**Petar Veličković** [1]

## Abstract

Verbal confidence—prompting LLMs to state their confidence as a number or category—is widely used to extract uncertainty estimates from black-box models. However, how LLMs internally generate such scores remains unknown. We address two questions: first, *when* confidence is computed – just-in-time when requested, or automatically during answer generation and cached for later retrieval; and second, *what* verbal confidence represents – token log-probabilities, or a richer evaluation of answer quality? Focusing on Gemma 3 27B (across TriviaQA, BigMath, and MMLU), Qwen 2.5 7B, and the reasoning model Magistral Small 24B, we provide convergent evidence for cached retrieval. Activation steering, patching, noising, and swap experiments reveal that confidence representations emerge at answer-adjacent positions before appearing at the verbalization site. Attention blocking pinpoints the information flow: confidence is gathered from answer tokens, cached at the first post-answer position, then retrieved for output. Critically, linear probing and variance partitioning reveal that these cached representations explain substantial variance in verbal confidence beyond token log-probabilities, suggesting a richer answer-quality evaluation rather than a simple fluency readout. These findings demonstrate that verbal confidence reflects automatic, sophisticated self-evaluation—not post-hoc reconstruction—with implications for understanding metacognition in LLMs and improving calibration.

## 1. Introduction

Confidence—a model's estimate that its answer is correct ([Pouget et al., 2016](#))—is critical to LLM reliability . Several

[1]Google DeepMind. Correspondence to: Dharshan Kumaran <dkumaran@google.com>.

*Proceedings of the 43$^{rd}$ International Conference on Machine Learning*, Seoul, South Korea. PMLR 306, 2026. Copyright 2026 by the author(s).

approaches have been developed to extract confidence estimates from LLMs. Token likelihoods yield well-calibrated confidences in multiple-choice or yes/no settings ([Kadavath et al., 2022](#); [Steyvers et al., 2025b](#); [Steyvers & Peters, 2025](#)); sampling-based consistency methods offer an alternative by measuring agreement across multiple model outputs ([Geng et al., 2023](#); [Tian et al., 2023](#)). However, most deployed models are black-box systems that do not expose token-level probabilities, limiting the applicability of these methods. This has motivated research into obtaining *verbal confidence* measures, where models are explicitly prompted to state their confidence as a number or confidence class (e.g., "Almost certain") ([Xiong et al., 2023](#); [Geng et al., 2023](#); [Yoon et al., 2025](#); [Steyvers et al., 2025a;b](#)).

Despite growing interest in verbal confidence as a practical measure of LLM uncertainty, little is understood about how such scores are internally generated. Understanding how LLMs compute confidence matters for several reasons. First, mechanistic insight could reveal whether verbal confidence reflects a meaningful self-evaluation of answer quality or merely surface-level correlates of generation fluency. This could thereby shed light on the metacognitive capacities of LLMs, and the observed difference in calibration between explicit verbal reports and token-level likelihoods ([Xiong et al., 2023](#); [Steyvers et al., 2025a](#)). Second, understanding these mechanisms could enable principled interventions to improve calibration, moving beyond prompt engineering or generic fine-tuning.

We ask two related questions. Our first question concerns *when* verbal confidence is computed—only when explicitly requested (i.e. at the last token of the prompt; a colon – see Figure [1](#)), or automatically during answer generation before the model knows a confidence rating will be required (i.e. at answer-adjacent tokens; the newline token in Figure [1](#)). Under the **just-in-time (JIT) hypothesis**, no dedicated confidence computation occurs during answer generation; instead, the model computes confidence only when prompted to self-evaluate, integrating features of the question and answer at that moment. Under the **cached retrieval hypothesis**, however, confidence is computed automatically as the answer is produced and stored for later retrieval when verbalization is required. Our second question asks *what* is verbal confidence—a readout of generation fluency and

token log-probabilities, or a richer evaluation of question-answer fit?

These questions parallel a longstanding debate in decision neuroscience concerning first-order versus second-order accounts of confidence (Fleming & Daw, 2017; Kepecs et al., 2008; Kiani & Shadlen, 2009)(see §A). Under first-order accounts, confidence arises from the same internal signals that drive the decision itself—confidence is a direct read-out of decision-variable strength. In LLMs, a first-order account would hold that verbal confidence is simply a read-out of token log-probabilities: the same signal that determined which answer tokens to generate also determines confidence. Under second-order accounts, confidence involves a distinct computation that evaluates the decision, drawing on partially independent signals. For LLMs, this would mean verbal confidence reflects something richer than log-probabilities—an evaluation of question-answer fit that goes beyond generation fluency. A key empirical implication is that second-order architectures can support error detection—recognizing that a response may be wrong even after committing to it—whereas pure first-order architectures cannot, because confidence and decision accuracy are yoked to the same underlying signal.

**Conflict of Interest Disclosure.** All authors of this paper are employed by Google DeepMind, which leads the development of the Gemma family of models. Gemma 3 27B is one of the three open-weight models evaluated in this work.

## 2. Experiments

We focus our experiments on Gemma 3 27B (Team et al., 2025) – given that this model allows access to the internal representations – and a prompt that asks for a class-based confidence report (Yoon et al., 2025) (see §B for full prompt used). We also report results with a prompt asking for a numeric confidence score (see Figure 15), Qwen 2.5 7b, and the reasoning model Magistral Small 24B. We primarily use the TriviaQA dataset which tests factual knowledge (Joshi et al., 2017), but also test the BigMath (Albalak et al., 2025) and MMLU (Hendrycks et al., 2021) datasets. Figure 10 provides a roadmap of the six experiments that follow, the specific question each addresses, and the key result from each.

Given the differing goals of our paper in comparison to that of Yoon et al. (2025), we deliberately suppress chain-of-thought reasoning in the main experiments by instructing the model to output only a confidence classification. This setting is practically relevant: many applications use LLMs for auto-rating or grading tasks without chain-of-thought to reduce costs. Moreover, even in reasoning models that externalize their thought process, decisions such as backtracking must be driven by latent signals computed within the forward pass

rather than derived from externalized tokens (Venhoff et al., 2025). Understanding how such internal representations are formed and accessed is therefore relevant beyond the no-CoT setting we focus on here. Our simplified setup allows us to more directly trace how confidence information flows through the network. Gemma was reasonably well calibrated (ECE = 0.12, AUROC = 0.71), and used both ends of the confidence class spectrum with reasonable frequency (though weighted towards higher confidence answers: see Figure 9).

### 2.1. Activation Steering

We first focus our attention on *when* – that is at what token position, and at which layers – confidence is represented in the model. If verbal confidence reflects access to meaningful internal uncertainty signals, then those signals should be instantiated in the model's activation space. This motivates the use of activation steering as a mechanistic probe (Turner et al., 2023; Stolfo et al., 2024a; Panickssery et al., 2023; Hua et al., 2025; Rai et al., 2024). Prior work shows transformers encode abstract properties such as love or hate as linear directions in activation space (Turner et al., 2023), implying high- and low-confidence trials should differ along identifiable directions.

We apply this framework to the generation of verbal confidence reports. By extracting confidence-encoding directions at different layers and token positions, and assessing how strongly they modulate expressed confidence when applied, we can track where confidence-relevant information is present in the network (see §C.1.3 for methodological details). Notably, we ruled out the possibility that steering vectors capture linguistic hedging rather than confidence per se, by verifying using GPT-4o-mini that none of the answers in our dataset – which were typically a few tokens in length – contained hedging language (e.g., "maybe", "probably", "perhaps").

Activation steering, therefore, allows us to distinguish between competing accounts of confidence generation. Under the just-in-time account, no confidence-specific computation occurs until the last token in the prompt (i.e. CC token—see Figure 1). Steering at the answer-adjacent token PANL should therefore be ineffective—PANL may carry answer-related information, but not a dedicated confidence representation that can be directly modulated. This follows because causal attention prevents PANL from attending to tokens that appear later in the prompt, so the model cannot "know" at PANL that a confidence rating will be requested. The cached retrieval hypothesis, however, holds that confidence is already encoded at PANL; steering should modulate this representation and bias downstream output. Note that both accounts predict that steering at CC can influence confidence, but only the cached account predicts that (1) steering

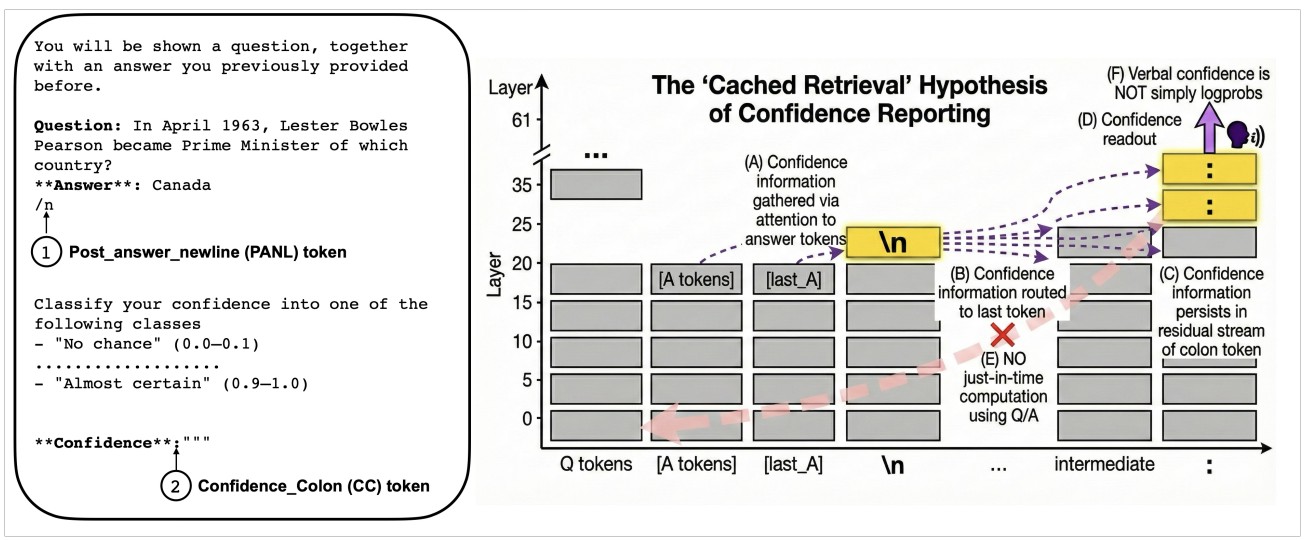

*Figure 1.* Main Prompt and Illustration of our findings. We included the generated answer (example question shown) from a previous phase as part of the prompt for the confidence rating experiment (see §C.1.2). Since the Transformer's forward pass is a function of previous tokens, providing the answer as context yields the exact same representation at the PANL as autoregressive generation. See §B for full prompt used. We provide convergent evidence that LLMs compute confidence via cached retrieval rather than just-in-time computation, and that verbal confidence doesn't merely reflect logprobs. (A) Confidence information is gathered at the post-answer-newline token (\n, PANL) via attention to answer tokens, particularly the last answer token, at earlier layers (21–25). (B) This information is routed to the confidence-colon token (:)—either directly or through intermediate tokens. (C) Confidence information persists in the residual stream at the confidence-colon through later layers (30–35). (D) Confidence is verbalized when CC's representation is transformed by the unembedding matrix at the final layer (layer 61). (E) Attention blocking experiments rule out just-in-time (JIT) computation: CC does not compute confidence from scratch by attending to question or answer tokens (red arrow from CC to Q and A tokens). (F) Decoding experiments reveal that verbal confidence is *not* explained by token log-probabilities, suggesting they reflect a more sophisticated evaluation of question-answer fit.

at PANL is effective and (2) PANL effects emerge at earlier layers than CC effects, reflecting the sequential flow from confidence encoding to retrieval.

We found evidence consistent with the cached retrieval account. High- (or low-) confidence steering caused a marked boost (or depression) in verbal confidence ratings when vectors were injected at PANL, with efficacy peaking at layers 21-25 (see Figure 2). Substantial steering efficacy was also observed at CC, peaking later at layers 30-35 —consistent with the prediction that automatically-formed confidence representations at PANL are subsequently transferred to CC for verbalization. As expected, steering was ineffective at control positions: PANL+1 and the first confidence colon token. Steering at the *answer-colon* (AC) position — the last token before answer generation, whose residual stream produces the first answer token's logits — was also ineffective (see later and Figure 13a), consistent with the view that verbal confidence is driven by an evaluation of question-answer fit rather than by the signals that produce the answer itself.

Steering at the first answer token was ineffective. Steering at the last answer token was effective, consistent with

confidence-relevant information being available at this point (i.e., after encoding the full question and most of the answer)(see Figure 11). However, we do not consider this position in subsequent analyses: unlike PANL, which immediately follows the answer, the last answer token is itself part of the answer content. This conflates two distinct roles—as a potential locus of cached confidence representations, and as part of the semantic content (answer correctness) from which confidence is presumably derived—introducing a confound that complicates interpretation.

## 2.2. Activation Patching

Having established through activation steering experiments that PANL and CC are key points where confidence is represented in the model, we next tested whether these positions are *sufficient* to restore confidence when the model's ability to assess answer quality has been disrupted.

To do this, we employed activation patching with a corrupt-then-restore paradigm (Meng et al., 2022; Heimersheim & Nanda, 2024; Zhang & Nanda, 2023)(see §A for more detailed literature review). First, we disrupted the model's access to answer information by replacing all answer token

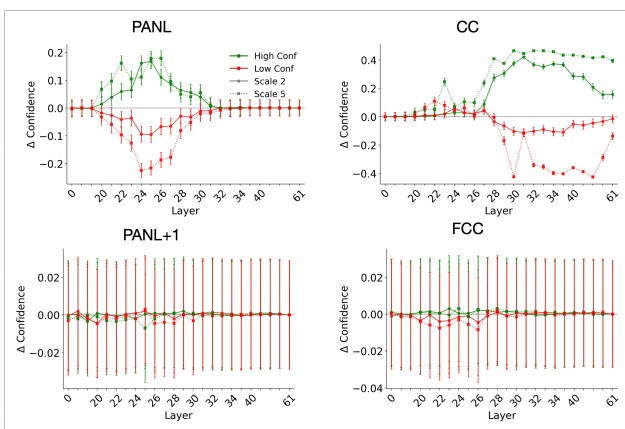

*Figure 2.* Results of Activation Steering In Gemma 3 27B. High (green lines) and low confidence (red lines) steering, at scales of 2 (solid line) and 5 (dotted line). Key positions: PANL (post-answer-newline) token, CC (confidence-colon) token. Control positions: PANL+1 (token immediately after PANL), FCC (first-confidence-colon) token (i.e. token preceding "$CLASS" in the prompt, following the confidence instructions; see Figure 8). Baseline confidence was 0.55 across all trials. See Error bars show SEM (n=200 trials).

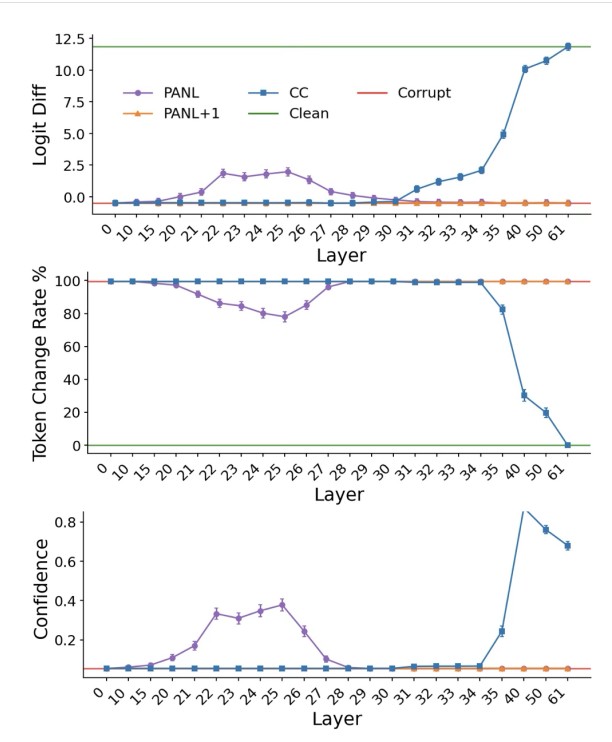

*Figure 3.* Results of Activation Patching in High Confidence Trials: Confidence Class Prompt. Clean baseline shown in green; corrupt baseline shown in red (i.e. at near zero for logit difference and confidence, and near 100 for first token change rate). Patching of PANL representation resulted in partial recovery of logit difference, first token and confidence (upper, middle, lower panel respectively). Patching of CC representation resulted in near complete recovery of confidence, logit difference and first token. PANL+1 patching resulting in effectively zero recovery.

activations with mean activations computed from 100 separate calibration trials (see §C.1.4 for methods). This corruption was applied at the input embedding level, propagating through all subsequent layers and effectively preventing the model from assessing whether its answer was correct. We restricted analysis to high-confidence trials because these provide the clearest test: corruption should substantially reduce confidence, whereas low-confidence trials are already near floor and thus less sensitive to disruption.

We first compared the corrupted condition to a clean run where answer tokens were untouched. We focus on three metrics throughout this and subsequent analyses (see §C.1.5 for details): (1) **confidence change**, the difference in reported confidence between conditions (mapping confidence classes to their numeric midpoints); (2) **logit difference change**, the change in the margin between the clean trial's confidence class logit and the mean logit of alternatives; we use this metric following (Wang et al., 2023; Heimersheim & Nanda, 2024; Rai et al., 2024), rather than probability because the model's computations are linear in logit-space until the final softmax; and (3) **first token change rate**, the proportion of trials where the argmax confidence token differed from baseline. Notably, we modified the prompt from Yoon et al. (2025) to ensure that each confidence class begins with a unique token, enabling meaningful analysis of metrics (2) and (3) (see Figure 8).

As expected, we found that during the corrupted run (i.e. after answer token corruption but without patching) the

model's confidence decreased to the lowest class, the difference between the logit of the first token outputted in the clean run and the mean of all other confidence classes collapsed to near zero, and there was a 100% change in the first token outputted by the model (see Figure 3).

We then selectively restored (i.e. patched) clean activations at at a single position (e.g. PANL) and at a single layer. The logic of this intervention is as follows: if a position at a given layer contains sufficient information to drive confidence output, then restoring clean activations there should recover the model's original confidence behavior despite the corrupted answer tokens propagating through the rest of the network. We tested three positions: PANL, CC, and PANL+1 as a control. Indeed, we found that patching of the CC and PANL representations – but not the PANL+1 representation – effected substantial recovery of all 3 metrics (confidence, logit difference, rate of change of first token).

Near-ceiling CC recovery at late layers is expected: CC's residual stream is directly transformed by the unembedding matrix, bypassing upstream corruption. More informative is the layer-wise pattern of recovery: PANL patching achieved

peak recovery earlier in the network (layer 25) compared to CC (which rose sharply only after layer 30). This temporal precedence is consistent with a cached retrieval mechanism in which confidence-relevant information is first consolidated at PANL before being transferred to CC for final output.

That PANL patching yielded only partial recovery is expected and consistent with broader findings in mechanistic interpretability. While some circuits can be cleanly isolated—such as the indirect object identification circuit (Wang et al., 2023)—recent work has shown that model behaviors typically arise from many overlapping heuristics rather than a single mechanism (Lindsey et al., 2025; Ameisen et al., 2025). Our intervention restored a single position at a single layer while answer token corruption continued to propagate through all other positions and layers; full recovery would require patching the complete distributed circuit. Furthermore, the pattern we observe is also consistent with cached retrieval being one of several overlapping mechanisms contributing to verbal confidence, rather than the sole computational pathway.

### 2.3. Activation Noising

To test whether PANL and CC representations are *necessary* for confidence reporting, we performed activation noising (mean ablation) experiments (Meng et al., 2022; Wang et al., 2023; Rai et al., 2024); see §C.1.6 for methods). For each position, we replaced its residual stream activation with the mean activation computed from a balanced set of 50 high-confidence and 50 low-confidence trials. We focussed on the two metrics that best capture disruption to the model's confidence-reporting mechanism: firstly the change in logit difference between clean and noised run, and secondly the first token change rate induced by noising.

We observed that mean ablation of PANL (peak layer 25) and CC (rising after layer 30) causes partial disruption of verbal confidence reporting, whilst no effect was observed at the PANL+1 control position (See Figure 12). This is consistent with the necessity of PANL and CC representations for faithful verbal confidence reporting. The partial rather than complete disruption may reflect either the distributed nature of confidence encoding across layers—such that ablating a single layer leaves other layers' contributions intact—or functional redundancy whereby alternative computational pathways can partially compensate for the ablated representation.

### 2.4. Activation Swap Experiment

The preceding experiments provide convergent evidence that PANL and CC carry confidence representations: steering vectors extracted from both positions produce graded, bidirectional modulation of confidence reports; patching

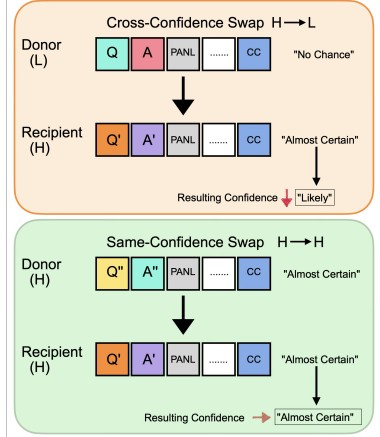

*Figure 4.* Illustration of Activation Swap Experiment. Upper panel: High→Low (i.e. cross-confidence swap: high confidence recipient trial receives low confidence donor representation) – result is a lowering of confidence. Lower panel: High→High (i.e. high confidence recipient trial receives High confidence donor representation) – in this same-same confidence swap, the result is no change in confidence.

partially restores confidence after answer corruption; and noising either position partially disrupts confidence output. Critically, effects at PANL emerge at earlier layers than at CC across all three experiments—consistent with the cached retrieval hypothesis in which confidence is first encoded at PANL and subsequently transferred to CC for verbalization. Control positions (PANL+1, FCC) – and the position directly involved in answer generation (Figure 13) – show no such effects.

To rule out the alternative that PANL encodes answer content rather than confidence, we designed an activation swap experiment (see Figure 4 and §C.1.7 for methods). This approach draws on the logic of interchange intervention (Geiger et al., 2021): if PANL caches a confidence representation, then transplanting PANL residual stream activations from a low (or high) confidence donor trial into a high (or low) confidence recipient trial in a *cross-confidence* swap trial should systematically bias the recipient toward the donor trial's confidence level—even though the recipient's question and answer remain unchanged. By contrast, if PANL primarily encodes content-specific features, then such cross-confidence swaps should induce only generic disruption effects that do not depend on the donor's confidence and are of similar magnitude to those observed in *same-confidence* control swap trials.

We observe that cross-confidence swaps at PANL (High→Low i.e. high confidence recipient trial receives low confidence donor representation) and (Low→High) produce systematic directional shifts in confidence—decreasing and increasing reported confidence, respectively—beyond the baseline disruption observed in same-confidence controls (see Figure 5). These effects peak at layer 26 and

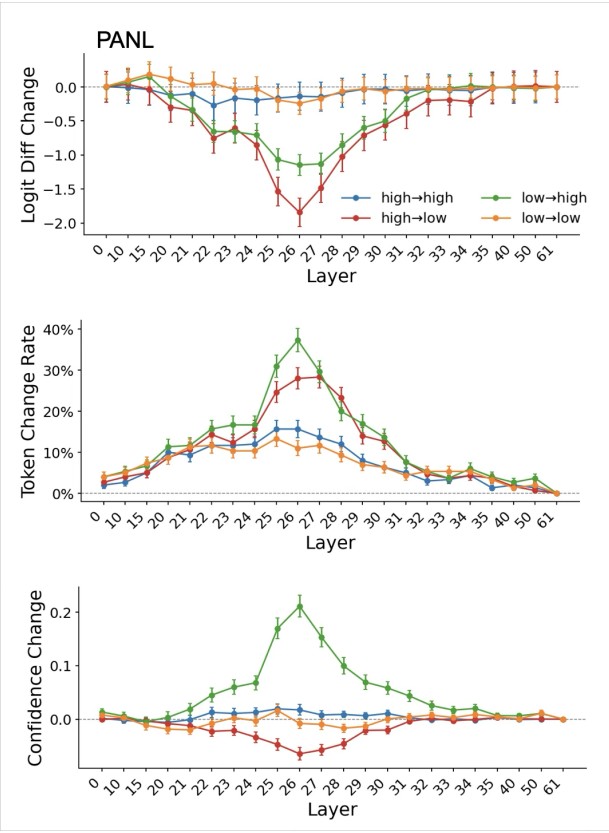

*Figure 5.* Results of Activation Swap Experiment at PANL position. Same-confidence swaps (H→H, L→L) control for generic content-related effects due to introducing activations from a different trial; cross-confidence swaps (H→L, L→H) isolate confidence-specific transfer. See main text for details. See Figure 20 for results at CC and PANL+1 positions.

are evident across all three metrics: first token change rate, logit difference change, and confidence change. The same pattern emerges at CC but at later layers (see Figure 20), consistent with the temporal precedence observed in previous experiments. No such effects are apparent at the PANL+1 control position (see Figure 20). These results provide compelling evidence that PANL representations carry confidence-specific information that transfers across trials—ruling out the possibility that PANL merely encodes content features that correlate with confidence.

### 2.5. Causal Interventions Remain Within the Natural Activation Distribution

We considered the possibility that causal interventions might push the model out-of-distribution producing generic effects. We found no evidence for this: across all four interventions, the OOD shifts in residual stream activations remained within the natural pairwise variability observed across trials, and were comparable at PANL and the control PANL+1 position — ruling out the alternative that PANL effects reflect generic out-of-distribution artifacts (Appendix D.4).

### 2.6. Decoding Confidence Information

The preceding experiments establish *when* and *where* confidence information is represented in the network. Specifically, we show that PANL and CC play causal roles in verbal confidence generation, with PANL's influence emerging earlier in the network – supporting the cached retrieval hypothesis. However, these interventions do not directly reveal *what* information is encoded at these positions.

While causal interventions test sufficiency/necessity, linear probing reveals where information first becomes decodable. We address two questions. First, is confidence information decodable from PANL at earlier layers than CC, as predicted by the cached retrieval hypothesis? Second, what type of confidence signal is encoded—do PANL representations simply summarize token log-probabilities (a white-box confidence measure that prior work has found to be better calibrated than verbal reports; Steyvers et al., 2025a; Kadavath et al., 2022), or do they reflect a distinct computation?

To address these questions, we trained linear probes to decode two measures from residual stream activations: answer correctness (binary classification, evaluated via AUROC) and verbal confidence magnitude (continuous regression using class midpoints, evaluated via $R^2$)(see §C.1.8 for methods). We additionally performed variance partitioning to assess whether PANL activations explain confidence variance beyond what is accounted for by log-probabilities associated with the actual answer tokens —as would be expected if PANL representations reflect a distinct computation rather than a simple summary of token likelihoods.

Consistent with the cached retrieval hypothesis, information about both correctness and confidence magnitude is decodable from PANL at earlier layers than from CC (see Figure 6, top and middle panels). Critically, variance partitioning revealed that activations at PANL and CC explain substantial unique variance beyond log-probabilities, confirming that confidence representations at these positions are not reducible to token probability signals (see Figure 6, bottom panel). Further, a combined log-probability baselines incorporating 6 indices (including length-normalized answer mean log-probability) explained only 10.0% of variance in verbal confidence ($r = 0.32$, $R^2_{CV} = 0.10$; see Figure 14, see D.3 and C.1.9 for details) – whereas PANL activation at L40 explained 38.0% variance. As expected, length-normalized mean answer-log-prob was a strong indicator of correctness (logprob AUROC = 0.75; see Figure 6) – as would be expected based on their widespread useage as a measure of confidence. These findings — together with the null effect at the answer-generation (AC) position (Figure 13) — suggest that verbal confidence reflects a distinct evaluation of question-answer fit, consistent with second-order models of confidence in neuroscience (see §A).

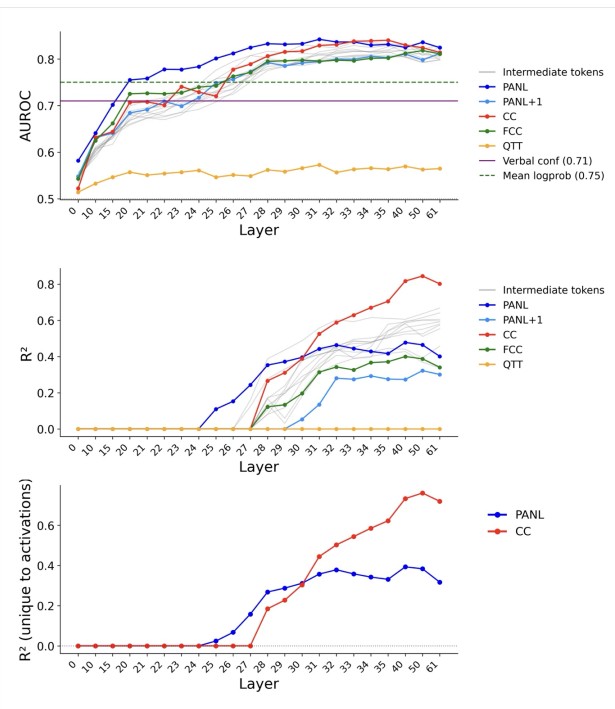

*Figure 6.* Results of decoding: linear probes were trained on residual stream activations at PANL, PANL+1, CC, intermediate tokens (first token of each confidence class in the instruction), and a control position (third question token). For binary correctness we report AUROC; for verbal confidence magnitude we report $R^2$. Horizontal lines show AUROC obtained using mean answer log-probability or verbal confidence directly. **Key results:** (1) confidence information is decodable from PANL at earlier layers than all other positions; (2) variance partitioning shows PANL activations explain substantial variance beyond log-probabilities ($R^2_{CV} = 0.084$ for logprobs alone), indicating PANL encodes information not captured by token likelihoods.

Our results also show that information about correctness (AUROC) and verbal confidence (R2) is distributed throughout the model – even at sites where steering (or patching type manipulations) has no effect (e.g. PANL+1, first-confidence colon; see Figure 2). This is consistent with previous studies showing information may be present in networks – and detected by decoding techniques – but not be observed in behavior (Azaria & Mitchell, 2023; Burns et al., 2022; Li et al., 2023; Liu et al., 2025; Bürger et al., 2024)(see §A for more detailed literature review). Furthermore, this finding aligns with prior work cautioning against over-interpretation of probing results without complementary causal interventions (Elazar et al., 2021; Ravichander et al., 2021).

### 2.7. The Answer-Colon Position is Not Causally Involved in Verbal Confidence Generation

A natural question is whether verbal confidence is generated through the same computational machinery that produces the answer itself. To address this, we examined the *answer-colon* (AC) token — the last token of the Phase 0 prompt,

immediately preceding answer generation. The AC token is causally implicated in answer generation, since its residual stream at the final layer is directly transformed by the unembedding matrix to produce the logits over the first answer token. Under a first-order account in which verbal confidence is a readout of token log-probabilities — the same signals that drove answer generation — interventions at AC should modulate verbal confidence. We found no evidence that this is the case: activation steering, patching, and noising at AC all produced null effects, indistinguishable from the PANL+1 control position, while the same interventions at PANL produced substantial effects on the same trials (Figure 13a–c). Linear decoding revealed weak residual signal at AC ($R^2 \sim 0.2$ across layers, compared to $R^2 \sim 0.75$ at PANL; Figure 13d). Because AC's residual stream directly produces the first answer token, this signal plausibly includes information about answer log-probability and other generation-time features. That this representation is both quantitatively weak *and* causally inert when the model verbalizes confidence is itself informative: the model has access to generation-time evidence at AC but does not primarily draw on it for verbal confidence reports. The richer, causally engaged confidence representation emerges later, at the post-answer position, supporting a second-order account (Fleming & Daw, 2017) in which verbal confidence reflects an evaluation of question-answer fit that is computationally distinct from the generation of the answer itself.

### 2.8. Attention Blocking Experiments

The preceding experiments establish that confidence information is encoded at PANL before CC, consistent with cached retrieval. However, three questions remain. First, can we exclude the possibility that just-in-time computation *also* occurs at the CC token through attention directed at question and answer tokens? Second, how does confidence information flow from PANL to CC, a process necessary for verbalization of this information? Third, how does confidence information reach PANL in the first place? Following Geva et al. (2023), we address these questions by selectively blocking attention edges between positions.

To address the first question, we blocked CC's attention to question and answer tokens (across all layers) using the categorical confidence prompt (see §C.1.10 for methods). This produced minimal effects ($\sim 10\%$ change rate), equivalent to a control condition blocking CC's attention to PANL+1— ruling out the just-in-time hypothesis that CC integrates question/answer information de novo (Figure 21). However, blocking CC→PANL also produced no effect with this prompt (Figure 21), suggesting that confidence information may flow through intermediate template tokens – of which there are more than a hundred in the categorical prompt – rather than via a direct pathway (see §D for details).

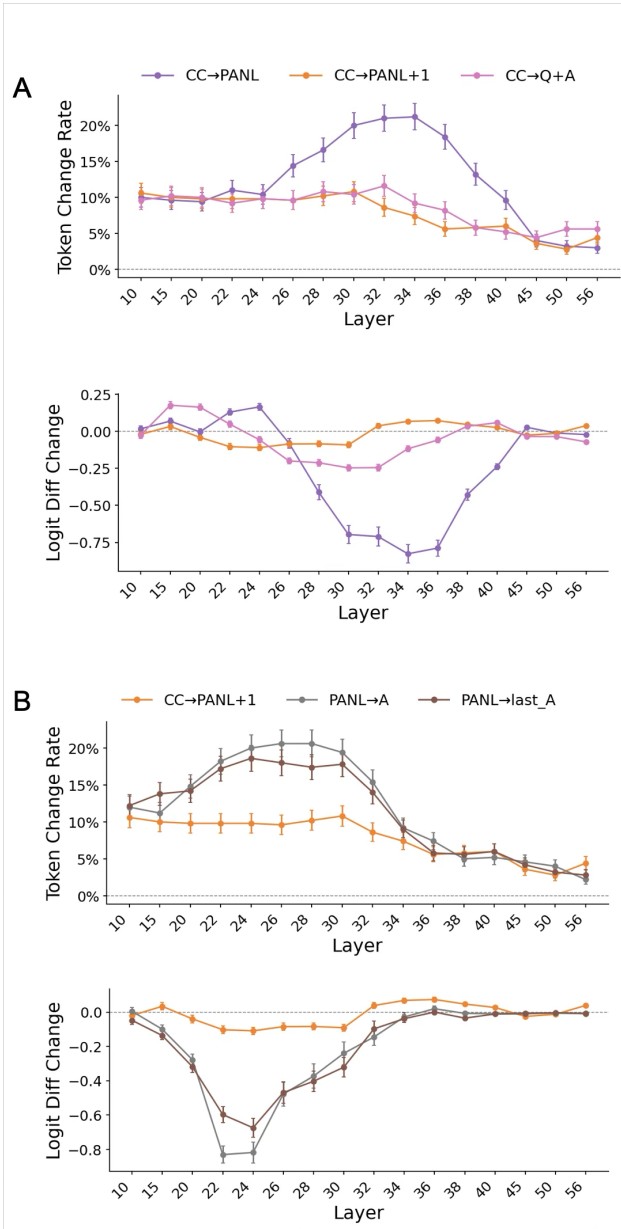

*Figure 7.* **Attention blocking reveals information flow from answer tokens through PANL to CC.** Results from selectively blocking attention edges across 12 consecutive layers centered at each x-axis position, using a minimal numeric (0–9) confidence prompt. **(A)** Blocking CC→PANL (purple) produces significant disruption peaking at layers 30–36, while blocking CC→Q+A (pink) shows no effect beyond the PANL+1 control (orange), ruling out just-in-time computation. **(B)** Blocking PANL's attention to answer tokens—either the last answer token (PANL→last_A, brown) or all answer tokens (PANL→A, gray)—produces disruption at earlier layers (22–28), preceding the CC→PANL effect. This temporal ordering indicates sequential information flow: answer tokens → PANL → CC. Upper panels show token change rate (%); lower panels show logit difference change (baseline token logit minus mean of alternative digits). Error bars indicate SEM ($n = 500$ trials).

To test whether CC retrieves information from PANL, we employed a pared down version of the numeric prompt that reduces the number of intermediate tokens between PANL and CC (see Figure 16). This prompt elicits numeric confidence (0–9) rather than categorical classes. Calibration was reasonable (ECE = 0.17; separate set of 2000 questions) and the model's confidence reliably discriminated correct from incorrect responses (AUROC = 0.68), confirming the presence of a meaningful confidence signal suitable for mechanistic analysis. We focus on analogous positions: PANL (the newline following the answer), PANL+1 (control), and CC (the final token before confidence generation) – and report the results using this minimal prompt below:

**CC does not compute confidence from scratch.** Blocking CC→Q+A produced effects indistinguishable from the PANL+1 control condition (∼10% change rate; Figure 7A), replicating the null result from the categorical prompt and confirming that CC does not perform just-in-time computation.

**CC retrieves confidence information from PANL.** In contrast to the categorical prompt, blocking CC→PANL in the minimal prompt produced substantial disruption: token change rates peaked at ∼21% (layers 30–36), accompanied by marked reductions in logit difference (Figure 7A). This effect significantly exceeded the PANL+1 control, providing direct evidence that CC retrieves confidence information from PANL when intermediate routing is minimized.

**PANL reads confidence from answer tokens.** To trace the upstream pathway, we blocked PANL's attention to answer tokens. Blocking PANL→last_A or PANL→A produced ∼20% token change rates peaking at earlier layers (22–28), with corresponding logit difference reductions (Figure 7B). These effects preceded the CC→PANL effects by approximately 6–8 layers, consistent with a sequential information flow: answer tokens → PANL → CC. See §D for details of complementary experiment with categorical prompt.

**Summary.** These attention blocking experiments rule out just-in-time computation and reveal the pathway through which confidence information flows: it originates at answer tokens, is read by PANL at earlier layers (22–28), and is subsequently retrieved by CC at later layers (30–36)(see Figure 1). The minimal prompt was essential for revealing the CC→PANL pathway, which was likely masked by redundant multi-hop routing through intermediate template tokens in the full categorical prompt.

### 2.9. Generalization across Prompt Format, Datasets, Architecture, and Reasoning Models

To assess generalization, we replicated our experiments across four axes: (1) a numeric (0–100) confidence prompt in Gemma 3 27B on TriviaQA (see Figures 15, 17 and 22);

(2) the categorical prompt in Qwen 2.5 7B (see §C.3); (3) two additional datasets — BigMath (Albalak et al., 2025) and MMLU (Hendrycks et al., 2021) — using Gemma 3 27B with the categorical prompt (see Figure 27 and Figure 28); and (4) a reasoning model with an extended chain-of-thought trace between question and final answer, Magistral Small 2506 (24B parameters, 40 layers) on TriviaQA (see Figure 25, Figure 26 §D.2 and §C.4).

The pattern of PANL playing a specific, causally sufficient role in verbal confidence generation — distinct from immediately adjacent control positions — held across all prompt formats, datasets, and the non-reasoning model architectures, with temporal precedence over CC where layer-wise analyses were performed (see Figures 22, 27, 28). In Magistral, the patching, swap, and decoding results similarly localized confidence representations to PANL, with the exception that distributed encoding of confidence across content-bearing trace tokens created redundant pathways that masked single-position noising effects at PANL (see §D.2). Together, these results indicate that the operation of cached retrieval reflects a general computational strategy for the generation of verbal confidence in LLMs rather than an artifact of prompt format, dataset, model architecture, or the absence of explicit chain-of-thought reasoning.

## 3. Related Work

We highlight work most relevant to our own here, and refer the reader to a detailed discussion of related work in the Appendix (see §A). Geva et al. (2023) showed that during factual recall, LLMs automatically cache attributes at the last subject token in early layers, later retrieving them for output rather than computing them de novo at the prediction site—a pattern strikingly similar to our cached retrieval hypothesis for confidence. Separately, classifiers trained on LLM hidden states distinguish true from false statements more accurately than methods based on output probabilities (Azaria & Mitchell, 2023; Burns et al., 2022), suggesting that models encode richer information about output quality than surface-level signals reveal—consistent with our finding that confidence representations contain information beyond token log-probabilities. Finally, Stolfo et al. (2024b) identified neurons in the final MLP layer that regulate confidence expression by modulating LayerNorm scale, but the upstream computation that produces the confidence signals these neurons act upon remained unexplored—a gap our work addresses.

## 4. Conclusion

Our results demonstrate that verbal confidence in non-reasoning and reasoning LLMs reflects cached retrieval rather than just-in-time computation. Addressing *when*

confidence is computed, we show that confidence representations emerge automatically during answer generation—before the model is aware a rating will be requested—rather than being constructed on-demand at verbalization. Addressing *what* confidence represents, we show that these cached representations explain substantial variance beyond token log-probabilities, and emerge at a position distinct from where the answer itself is generated — together suggesting a richer evaluation of question-answer fit consistent with a second-order model of confidence (Fleming & Daw, 2017) rather than a simple generation-related fluency readout. Consistent with broader findings that model behaviors typically arise from many overlapping heuristics rather than single mechanisms (Lindsey et al., 2025; Ameisen et al., 2025), we view cached retrieval as the dominant rather than the sole computational pathway in the settings we study; distributed or overlapping circuits may also contribute, particularly in regimes we have not tested.

The automatic computation of confidence alongside answer generation parallels findings in factual recall, where LLMs automatically enrich subject representations with many attributes during early layers before extracting the specific queried attribute at later layers (Geva et al., 2023). Subsequent work points to a role for a cached representation at the immediate post-answer position in error detection (Kumaran et al., 2026c), suggesting that the mechanism we characterize here is recruited for self-evaluation more broadly. Together, these results are consistent with recent evidence that LLMs possess some degree of introspective awareness (Anthropic, 2025), though whether the process we characterize constitutes introspection in a stronger sense remains an open question.

## Impact Statement

This work advances mechanistic understanding of how large language models generate verbal confidence reports. Such understanding may support more reliable uncertainty estimation by helping identify how confidence signals are formed and where they can become misaligned with answer correctness. This is important as LLMs are deployed in high-stakes settings where misleading confidence may distort user trust, and improved confidence estimation could contribute to detecting hallucinations and other model errors. As with reliability research more broadly, however, gains in model trustworthiness may also encourage deployment in higher-stakes contexts where residual failures and tail risks remain consequential.

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

# Appendix Overview

- **Appendix A: Related Work** (§A) — Summary of related literature

- **Appendix B: Supplemental Figures** (§B) — Prompts, calibration plots, and additional experimental results (including Magistral Small 24B)

- **Appendix C: Supplemental Methods** (§C)

  - C.1 Experiments with Categorical Confidence Prompt in Gemma 3 27B (§C.1)
    * C.1.1 Technical Details (§C.1.1)
    * C.1.2 Generation of Answers (§C.1.2)
    * C.1.3 Activation Steering (§C.1.3)
    * C.1.4 Activation Patching (§C.1.4)
    * C.1.5 Metrics (§C.1.5)
    * C.1.6 Activation Noising (§C.1.6)
    * C.1.7 Activation Swap (§C.1.7)
    * C.1.8 Decoding Confidence Information (§C.1.8)
    * C.1.9 Variance partitioning using log-probability baselines (§C.1.9)
    * C.1.10 Attention Blocking Method (§C.1.10)
  - C.2 Experiment using Numeric Confidence Prompt in Gemma 3 27B (§C.2)
  - C.3 Experiment using Qwen 2.5 7B (§C.3)
  - C.4 Experiment using Magistral Small 24B Reasoning (§C.4)

- **Appendix D: Supplemental Results** (§D)

  - D.1 Attention Blocking with Categorical Confidence Prompt (§D.1)
  - D.2 Experiments with Magistral Small 24B Reasoning (§D.2)
  - D.3 Gemma 3: Variance partitioning using log-probability baselines (§D.3)
  - D.4 Gemma 3: Causal Interventions Remain Within the Natural Activation Distribution (§D.4)
  - D.5 Limitations and Future Work (§D.5)

# A. Related Work

**Confidence and Calibration in LLMs.** Confidence refers to a model's estimate of the probability that its output is correct (Pouget et al., 2016), while calibration measures the alignment between expressed confidence and actual accuracy (Guo et al., 2017). Well-calibrated confidence estimates are critical for deploying LLMs in high-stakes applications and have may be useful for detecting hallucinations (Varshney et al., 2023). Several methods exist for eliciting confidence from LLMs: token-level likelihoods provide well-calibrated estimates, particularly for multiple-choice or yes/no verification tasks (Kadavath et al., 2022; Steyvers et al., 2025b; Steyvers & Peters, 2025), while sampling-based approaches assess confidence through consistency across multiple outputs (Geng et al., 2023; Tian et al., 2023). However, these white-box methods require access to internal probabilities unavailable in most deployed systems. This limitation has driven interest in *verbal confidence*, where models are prompted to explicitly report their confidence as a numeric value or categorical label (e.g., "Almost certain") (Xiong et al., 2023; Geng et al., 2023; Yoon et al., 2025; Steyvers et al., 2025a; Yang et al., 2024). The class-based prompt used in our main experiments is derived from Yoon et al. (2025) – modified so that each first token is unique to the class.

**Latent Representations of Uncertainty and Correctness.** A growing body of work demonstrates that LLMs encode information about the quality of their outputs within internal activations, often in ways that diverge from surface-level confidence. Teplica et al. (2025) find that the same model components contribute to both answer accuracy and decoded uncertainty estimates from P(IK) probes (Kadavath et al., 2022). Azaria & Mitchell (2023) showed that models encode a latent representation of truthfulness in their hidden states: classifiers trained on these activations distinguish true from false statements with 60–80% accuracy, generalizing across topics and outperforming prompting-based methods. Critically, classifiers trained on these internal activations outperform methods based on output probabilities, which are confounded by factors such as sentence length and token frequency and thus provide less reliable signals of truthfulness. Similarly, Liu et al. (2025) found that activations at the first output token predict answer correctness with approximately 75% accuracy, and introduced metrics to distinguish correct, incorrect, and irrelevant retrieval contexts directly from model internals. These findings align with broader evidence that task-relevant information may be decodable from network activations yet remain dissociated from observed behavior (Burns et al., 2022; Li et al., 2023; Bürger et al., 2024).

**Mechanistic Interpretability: Activation Steering** Activation steering is a technique for causally intervening on model behavior by adding or subtracting directions in activation space. Turner et al. (2023) demonstrated that abstract concepts such as love or hate are encoded as approximately linear directions in transformer representations. These directions can be recovered through various methods, including contrasting mean activations between conditions that differ along a conceptual dimension. Subsequent work has applied steering to modify instruction-following behavior (Stolfo et al., 2024a), confidence-guided abstention (Kumaran et al., 2026a), reasoning (Venhoff et al., 2025), sycophancy and other traits (Panickssery et al., 2023), and evaluation-aware responses (Hua et al., 2025).

**Activation Patching.** Activation Patching (also termed causal tracing) identifies which model components are causally responsible for specific behaviors. Meng et al. (2022) used the corrupt-then-restore paradigm: inputs are corrupted to disrupt model output, then clean activations are selectively restored at specific position-layer combinations to measure recovery. Positions that restore performance when patched are implicated as causally sufficient for the computation. This approach has been extended to study indirect object identification (Wang et al., 2023), and best practices for activation patching metrics have been systematically evaluated (Zhang & Nanda, 2023; Heimersheim & Nanda, 2024).

**Activation Noising.** Activation noising (or ablation) assesses the necessity of specific model components by removing their contribution during the forward pass and measuring the resulting impact on model behavior (Meng et al., 2022; Wang et al., 2023; Rai et al., 2024). To improve the reliability of these interventions, mean ablation replaces a component's activation with an average computed across multiple samples from the data distribution (Wang et al., 2023). This approach mitigates the out-of-distribution (OOD) effects associated with zero or random noising by ensuring the intervention remains grounded in valid, in-distribution model states (Zhang & Nanda, 2023).

**Interchange Intervention.** Geiger et al. (2021) formalize activation swapping as a rigorous test for causal abstraction: a neural representation can be shown to encode a specific high-level concept if swapping it systematically controls the model's output behavior. This framework provides the theoretical foundation for our activation swap experiments, establishing that such interventions are the standard method for demonstrating a causal link between an internal state and a high-level variable.

**Confidence Regulation Neurons.** Stolfo et al. (2024b) identify two classes of neurons in the final MLP layer that regulate output confidence: entropy neurons, which modulate the LayerNorm scale by writing to an effective null space of the unembedding matrix, and token frequency neurons, which shift the output distribution toward or away from the unigram distribution. Their analysis demonstrates that these components serve a calibration function, with entropy neurons acting as a hedging mechanism that increases uncertainty to mitigate loss spikes on incorrect predictions. This work provides evidence that focuses on output-level calibration rather than the computation of confidence representations during answer generation.

**Attention Blocking.** Geva et al. (2023) employ attention blocking to demonstrate that factual recall relies on a distinct cache-and-retrieve mechanism, where knowledge is aggregated and stored at the last subject token in early layers rather than being accessed directly at the output. They show that this token acts as a temporary holding state for factual attributes, which are subsequently retrieved by the final prediction token via specific attention heads in the network. This establishes that information flow is spatially localized, relying on intermediate tokens to compute and maintain latent states during the forward pass.

**Theories of Confidence in Decision Neuroscience.** Our investigation connects to a longstanding debate in decision neuroscience concerning the computational basis of confidence (Fleming & Daw, 2017; Pouget et al., 2016; Kepecs & Mainen, 2012; Kiani & Shadlen, 2009). Under first-order accounts, confidence arises from the same internal signals that drive the decision itself. In perceptual tasks, for instance, both the choice and confidence derive from a single decision variable representing accumulated evidence; confidence is simply a readout of how strongly this variable favored the chosen option. Translated to LLMs, a first-order account would hold that verbal confidence is a readout of token log-probabilities—the same signals that determined which answer tokens to generate also determine confidence.

Under second-order accounts, confidence involves signals that are distinct from—though correlated with—those driving the decision (Fleming & Daw, 2017). These additional signals enable an evaluation of the decision that goes beyond the information directly used to produce it. For LLMs, evidence that verbal confidence reflects information beyond token log-probabilities would suggest a second-order-like computation capable of genuine answer-quality evaluation. However, second-order signals need not be unbiased: in humans, post-decisional confidence is often inflated for the chosen option relative to alternatives, a pattern known as choice-supportive bias (Henkel & Mather, 2007). Recent work has documented analogous biases in LLMs, where confidence in an answer can be inflated by the prior act of choosing it, producing systematic overconfidence (Kumaran et al., 2026b).

An important consequence is that second-order architectures can support error detection (Fleming & Daw, 2017; Kumaran et al., 2026c): because confidence draws on partially independent information, the system can recognize that a response may be wrong even after committing to it. In contrast, pure first-order architectures cannot detect errors, because confidence and decision accuracy are yoked to the same underlying signal.

# B. Supplemental Figures

```
"""You will be shown a question, together with an answer you previously provided before.

Question: {question}
**Answer**: {answer}
\n  ←——————————  ①  post_answer_newline token
Classify your confidence into one of the following classes based on how
likely the answer above is to be correct (NO REASONING OR EXPLANATION):

- "No chance" (0.0—0.1)
- "Really unlikely" (0.1—0.2)
- "Chances are slight" (0.2—0.3)
- "Unlikely" (0.3—0.4)
- "Less than even" (0.4—0.5)
- "Better than even" (0.5—0.6)
- "Likely" (0.6—0.7)
- "Very good chance" (0.7—0.8)
- "Highly likely" (0.8—0.9)
- "Almost certain" (0.9—1.0)

Each category reflects the probability that the answer is correct.

At the very end of your output, format your confidence as
**Confidence**: $CLASS
where CLASS is one of the names (only the names without the probability ranges)
of the classes above.

**Confidence**:"""
                    ↑
              ②  confidence_colon token
```

*Figure 8.* Full categorical confidence class prompt in Experiment. We focused our analysis on the newline token following the model's answer (given in a previous phase): the post-answer-newline (PANL) token, and the confidence-colon token (i.e. the last token of the prompt). In addition, we report analyses on the token immediately following the PANL token (i.e. PANL-plus1 token), the first-confidence-colon (FCC)(i.e. the colon preceding the appearance of "$CLASS"), and the last token of the answer. This prompt is derived from (Yoon et al., 2025) but with the following key modification: the first token of every confidence class is unique, allowing us to meaningfully analyze changes in the ID of first token, the logit of the first token etc.

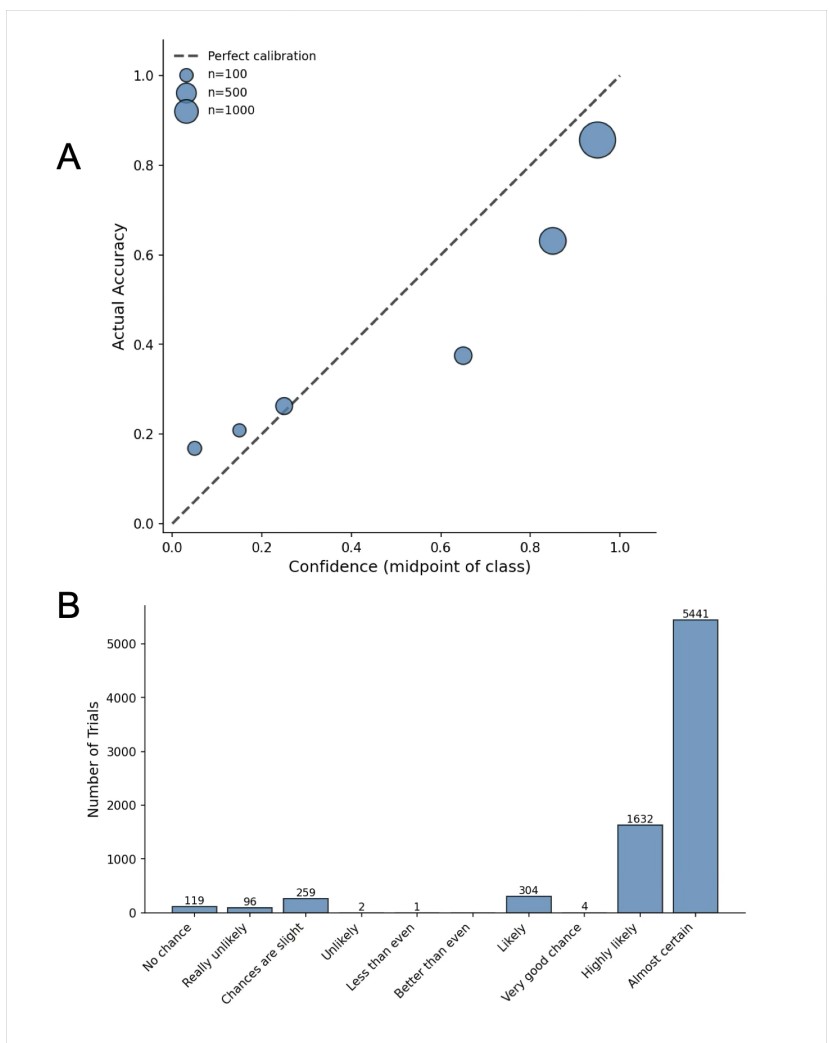

*Figure 9.* Calibration and Distribution of Confidence Classes in Gemma. (A) Calibration of Gemma: Expected Calibration Error (ECE) = 0.12, AUROC = 0.71. No procedures such as temperature scaling (Guo et al., 2017) were used, since we were focussed on understanding the generation of Gemma's raw verbal confidence signals. The model's performance was 77.4%; this was determined by having GPT4o-mini mark questions (B) Distribution of Gemma's confidence responses across the 10 classes. n = 7858 questions from the TriviaQA dataset (Joshi et al., 2017).

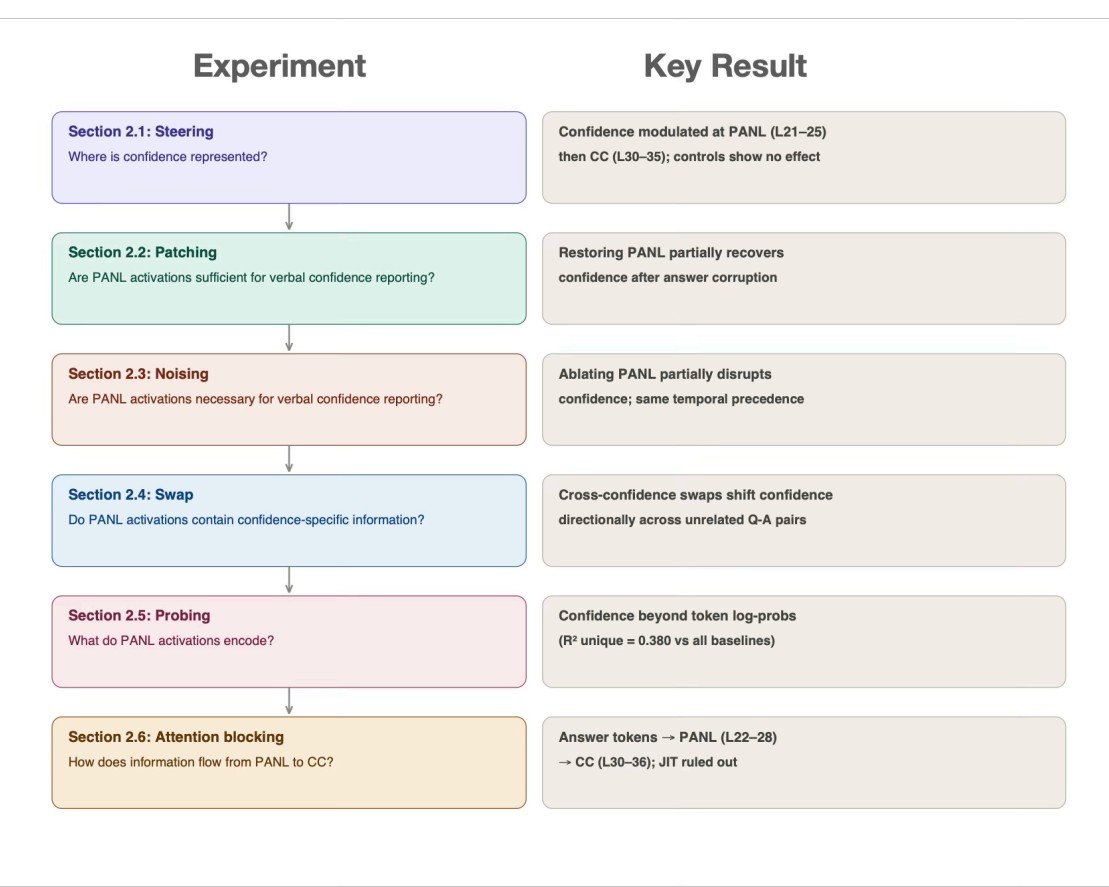

*Figure 10.* Sequence of Experiments, Motivation and Brief Summary of Key Results.

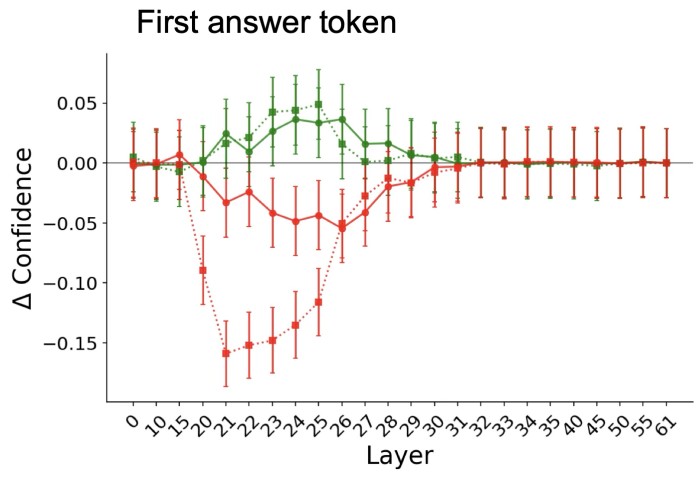

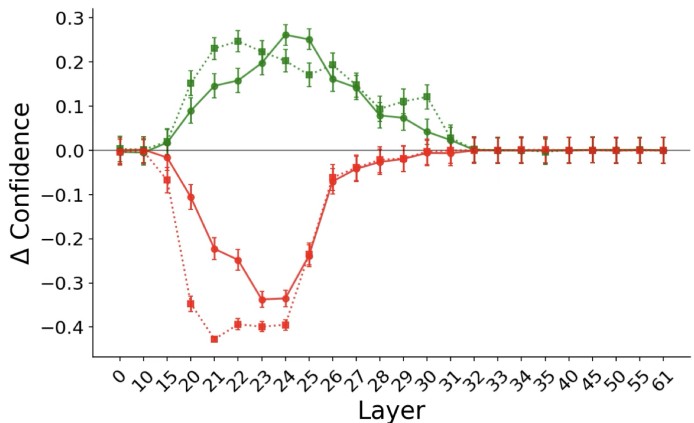

*Figure 11.* Results of Activation Steering in Gemma at answer tokens (Confidence Class Prompt). High (green lines) and low confidence (red lines) steering, at scales of 2 (solid line) and 5 (dotted line). Error bars show SEM.

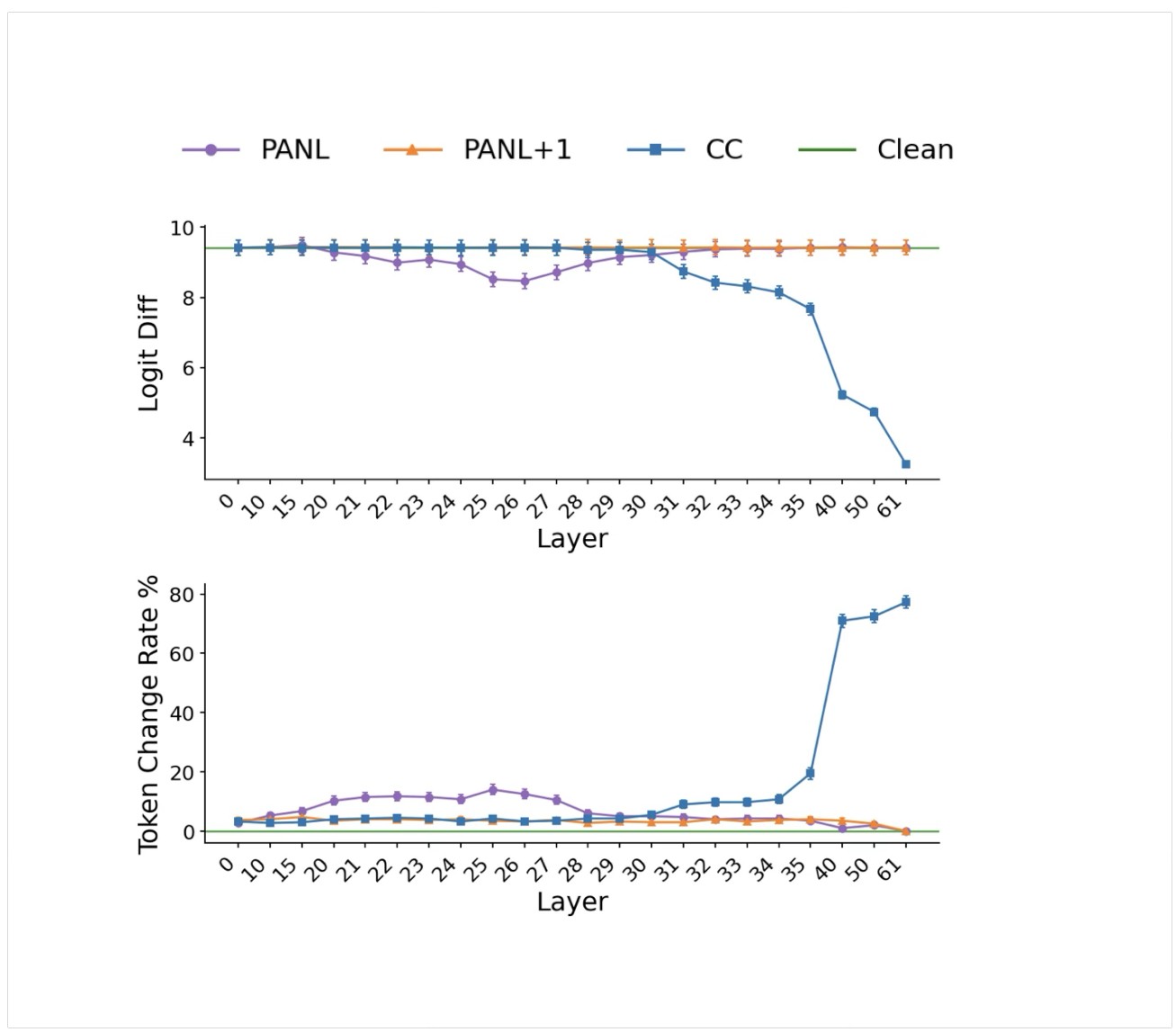

*Figure 12.* Results of Activation Noising across all trials. Mean ablation of representations of PANL or CC token at a single layer causes disruption of verbal confidence reporting as measured by decrease in logit difference, and change in first token outputted by model. See text for details

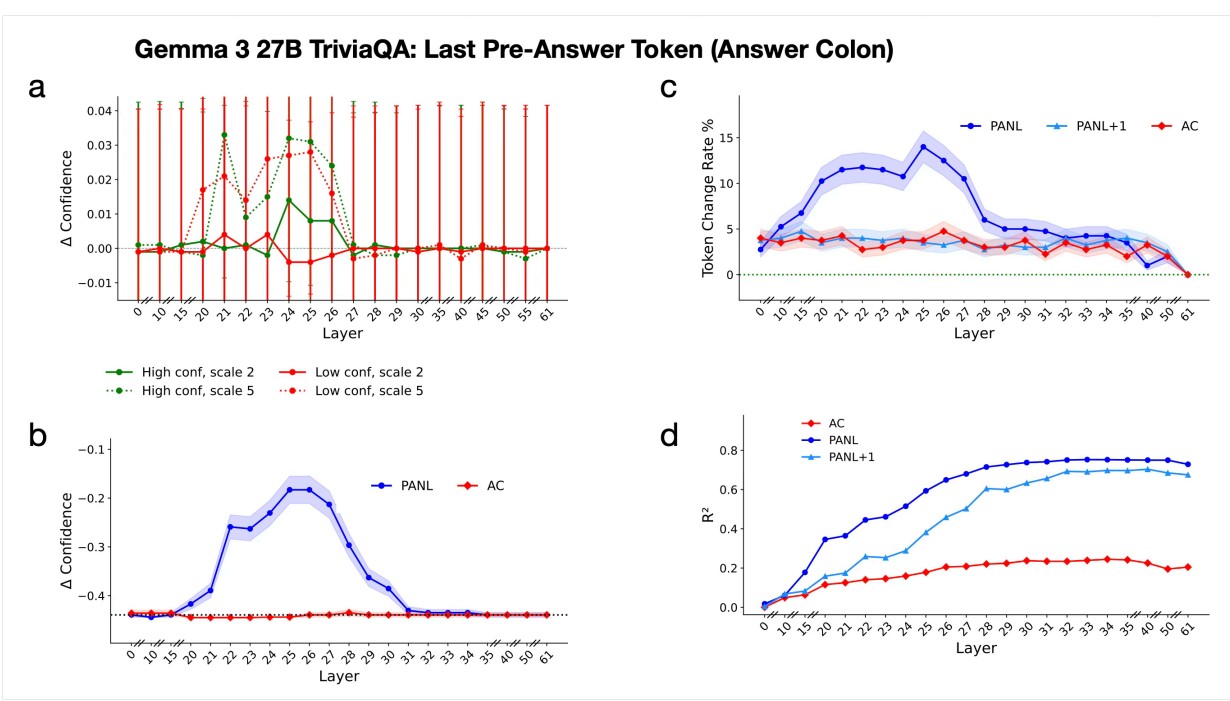

*Figure 13.* **Results relating to interventions at the last pre-answer position: the answer-colon (AC) token**. a) results of activation steering. No significant change in confidence. b) Activation patching: experiment performed by corrupting AC token (in addition to answer tokens). Confidence change from clean baseline shown. Partial recovery of confidence observed at PANL but no recovery at AC. c) Activation noising: no disruption of confidence reporting observed with mean ablation of AC position – i.e. result is comparable to control position PANL+1. d) Encoding of verbal confidence information: layerwise linear decodability of verbal confidence from residual activations at AC, PANL, and PANL+1, measured by cross-validated ridge regression $R^2$ – with best alpha (penalty strength) for AC position used also for PANL and PANL+1. Information related to verbal confidence is present at AC to a much lower extent that at other 2 positions (see Main text for interpretation)

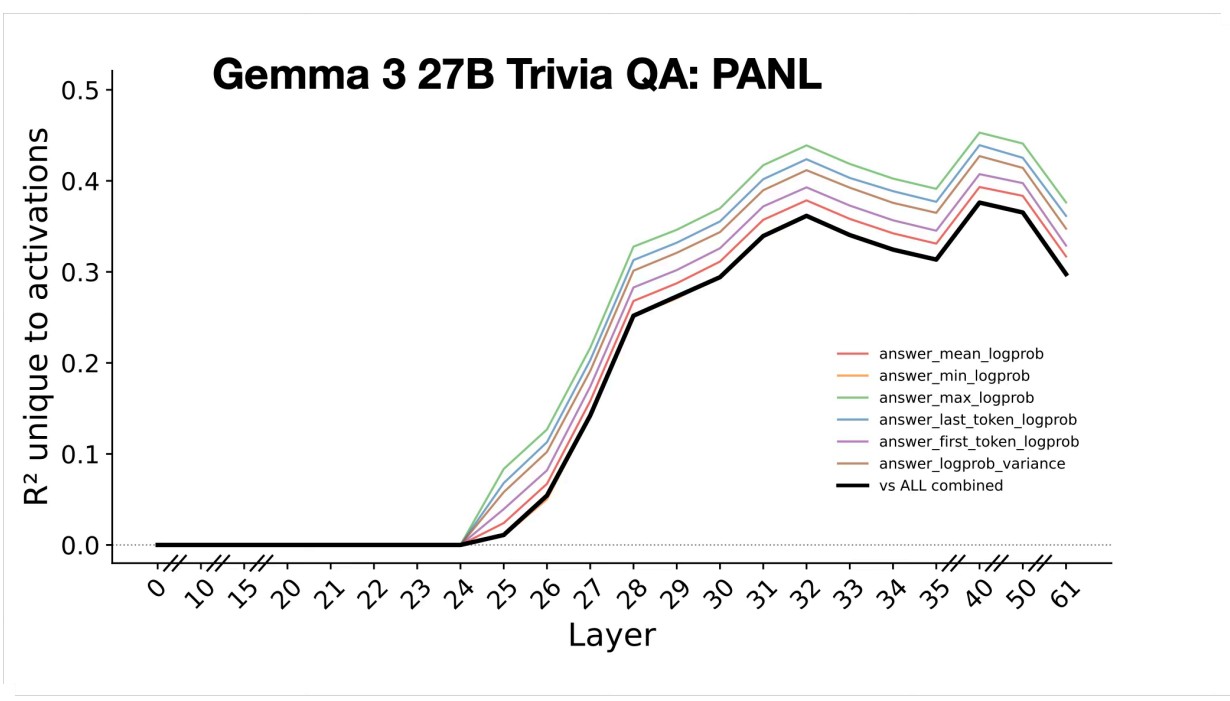

*Figure 14.* **Variance in verbal confidence uniquely explained by PANL activations '$R^2_{\text{unique}}$' after controlling for token-level probability baselines, plotted across layers.** Each colored line shows '$R^2_{\text{unique}}$' after controlling for a single baseline; the black line shows '$R^2_{\text{unique}}$' after controlling for all six baselines combined. PANL activations explain substantial unique variance ($R^2_{\text{unique}} = 0.38$ at peak layer ~40) beyond any combination of answer token log-probability summaries, confirming that the confidence representation at PANL is not reducible to token-level probability signals.

```
"""You will be shown a question, together with an answer
you previously provided before.
Question: {question}
**Answer**: {answer}

State your confidence as an integer between 0 and 100 based
on how likely your answer is to be correct.
That is, if your confidence is 0, that means that your
answer has almost no chance of being correct.
If your confidence is 100, then you are almost certain that your
answer is correct.

At the very end of your output, format your confidence as:
**Confidence**:$CONFIDENCE
where CONFIDENCE is an integer between 0 and 100.
do NOT leave a space before the first digit or you will be
scored WRONG!
**Confidence**:"""
```

*Figure 15.* Full numeric confidence prompt used in Experiment. This prompt is derived from (Mei et al., 2025; Devic et al., 2025)

```
"""You will be shown a question, together with an
answer you previously provided before.
Question: {question}
**Answer**: {answer}
\n  ←————————  (1)  post_answer_newline token
**State Confidence(0-9) with NO SPACE**:'"""
                                    ↑
                    confidence_colon token  (2)
```

*Figure 16.* Minimal numeric confidence prompt used in attention blocking experiments. This prompt elicits confidence on a 0–9 scale (single token output) and minimizes intermediate tokens between the post-answer-newline token (PANL, position 1) and the confidence-colon token (CC, position 2). These positions are analogous to those in the main categorical prompt (Figure 8). The minimal design enables direct testing of whether CC retrieves confidence information from PANL, a pathway that may be masked by routing through intermediate template tokens in the categorical prompt.

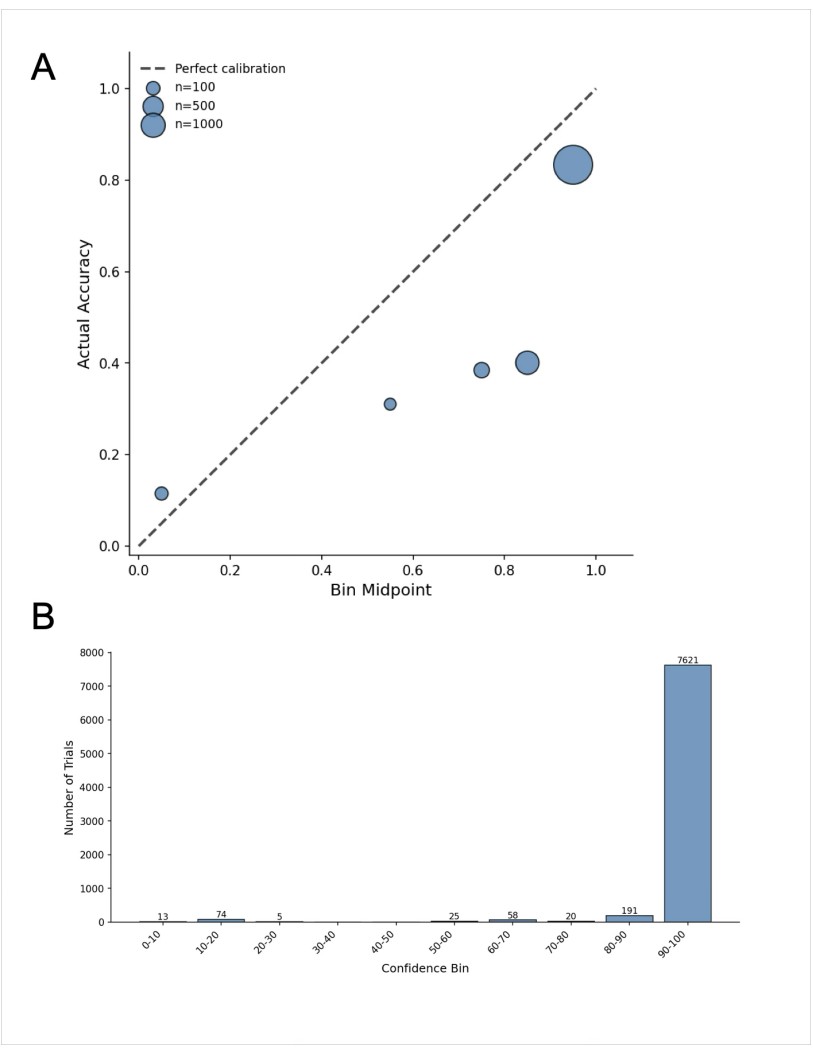

*Figure 17.* Calibration and Distribution of Numeric Confidence Scores in Gemma. (A) Calibration of Gemma: Expected Calibration Error (ECE) = 0.16, AUROC = 0.73. No procedures such as temperature scaling (Guo et al., 2017) were used, since we were focussed on understanding the generation of Gemma's raw verbal confidence signals. (B) Distribution of Gemma's numeric confidence responses binned into 10 bins. Questions (n = 8008) the TriviaQA dataset (Joshi et al., 2017).

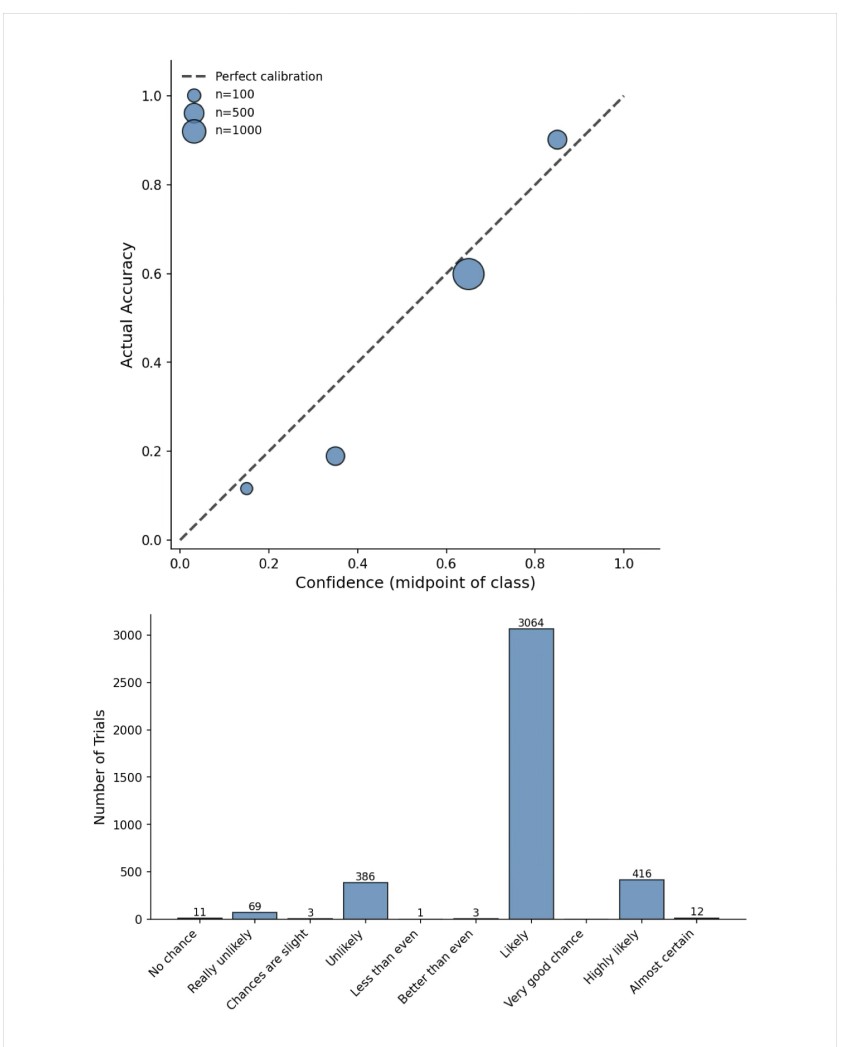

*Figure 18.* Calibration and Distribution of Categorical Confidence Ratings in Qwen 2.5 7b. (A) Calibration of Qwen: Expected Calibration Error (ECE) = 0.06, AUROC = 0.65. No procedures such as temperature scaling (Guo et al., 2017) were used, since we were focussed on understanding the generation of Qwen's raw verbal confidence signals. (B) Distribution of Qwen's confidence responses.

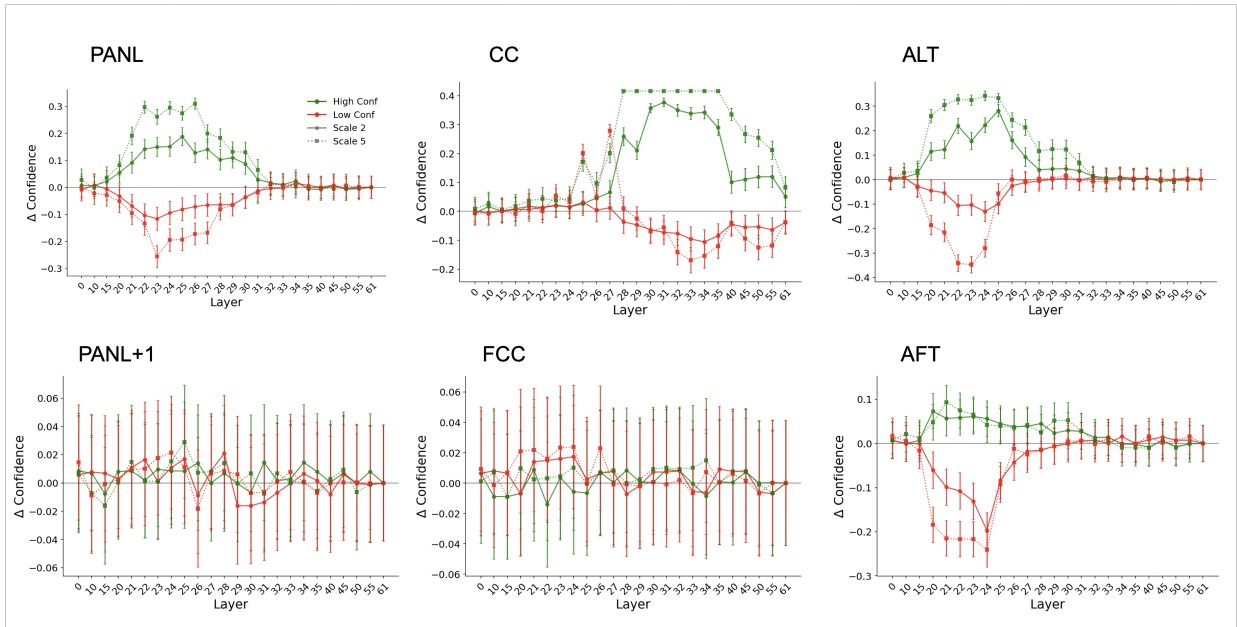

*Figure 19.* Results of Activation Steering in Gemma with Numeric Confidence Prompt. High (green lines) and low confidence (red lines) steering, at scales of 2 (solid line) and 5 (dotted line). n = 124 trials per condition per layer. Positions correspond to the analogous locations in the categorical confidence prompt. Error bars show SEM.

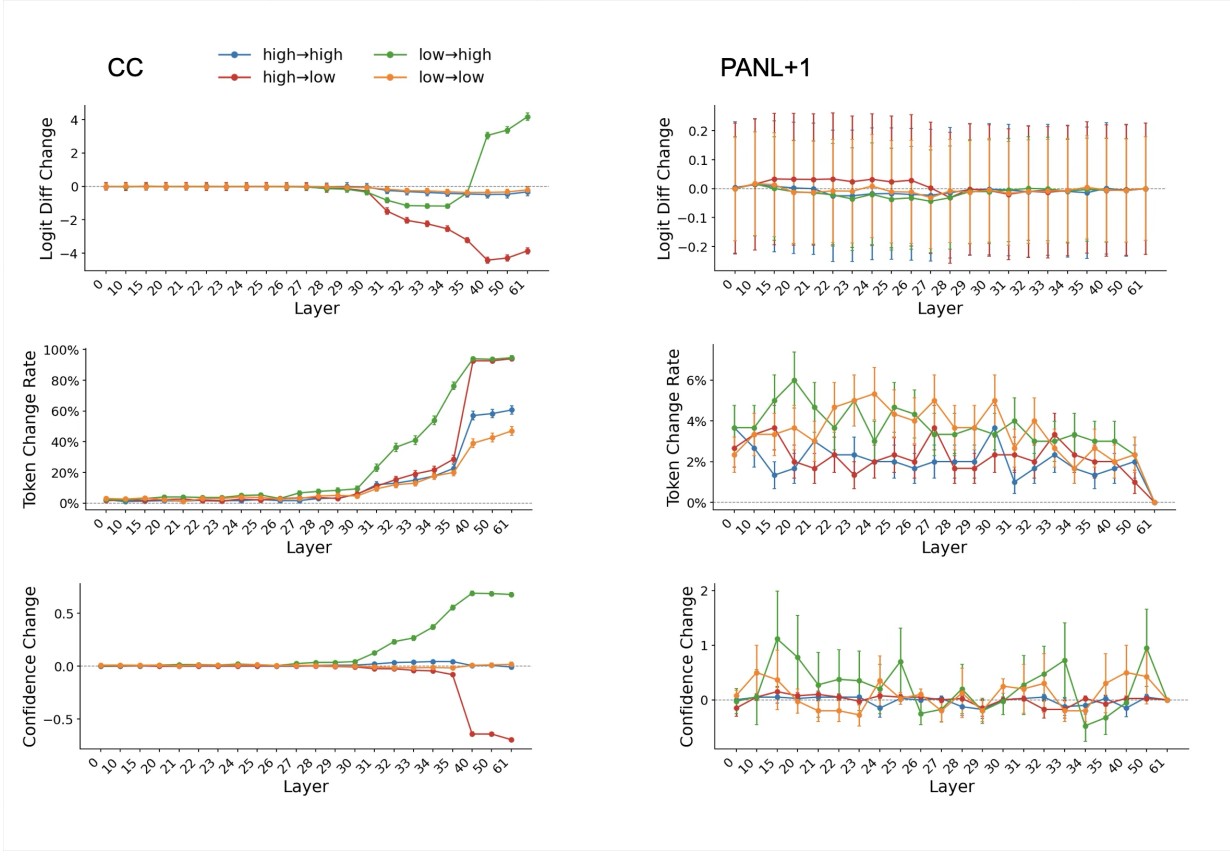

*Figure 20.* Results of Activation Swap Experiment at CC and PANL+1 position (Confidence Class Prompt). Same-confidence swaps (H→H, L→L) control for cross-trial substitution effects; cross-confidence swaps (H→L, L→H) isolate confidence-specific transfer. See main text for details.

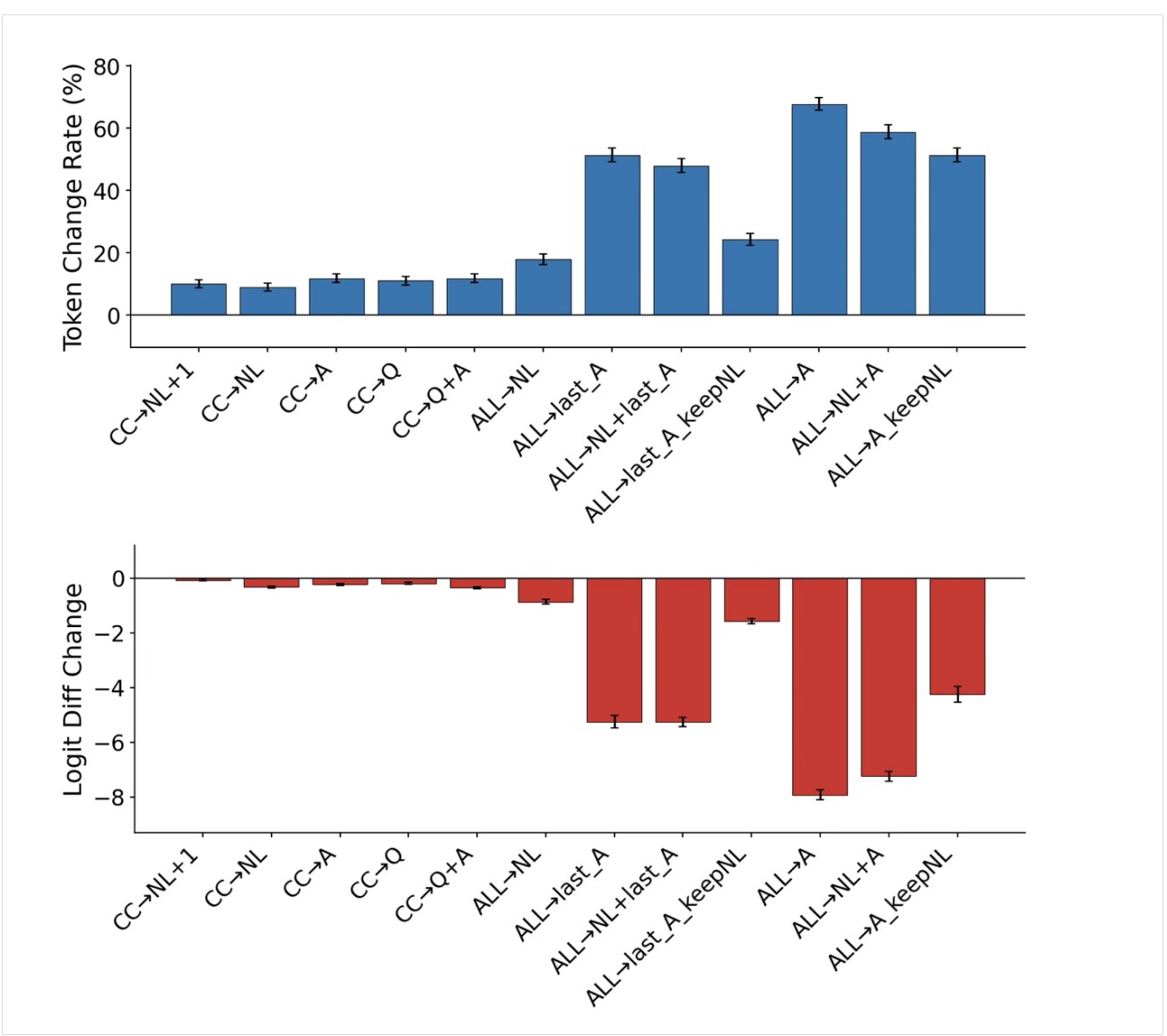

*Figure 21.* Results of Attention Blocking Experiment using Main Categorical Prompt. First token change rate induced by blocking different attentional pathways shown in upper panel. Lower panel: Change in logit difference between the first token and the mean of alternative confidence classes caused by attention blocking. PANL and PANL+1 denoted by NL and NL+1 for brevity. last_A and A refer to last answer token and answer tokens, respectively. Q refers to all question tokens. See text for interpretation.

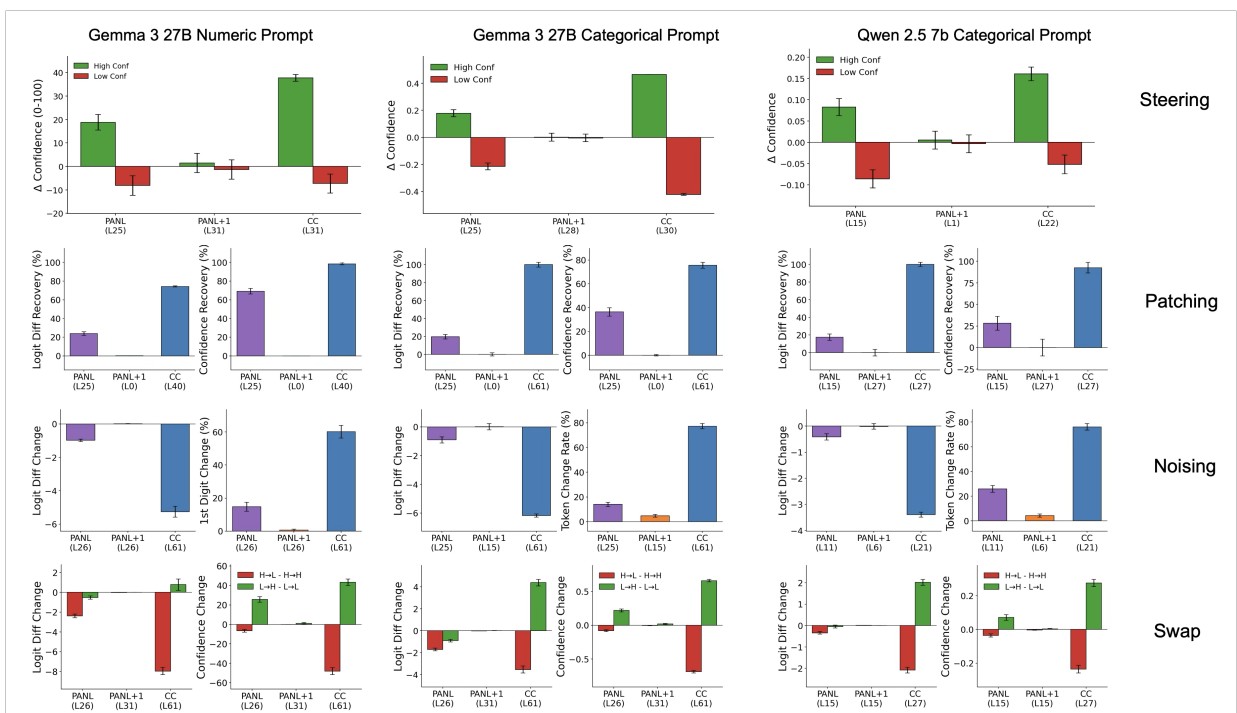

*Figure 22.* Results for Gemma 3 27B and Qwen 2.5 7B on Activation Steering, Patching, Noising and Swap Experiments: Categorical Confidence and Numeric Confidence Prompt. Maximal effects observed at each position's (PANL, PANL+1, CC) peak layer (layer index below bar). Qwen has 28 layers, Gemma 3 27B has 62 layers. Logit difference change (i.e. change from logit of clean run first token vs average of all other confidence classes), and confidence change reported; except for noising experiment where first token change rate and logit difference are reported as measures of confidence reporting disruption (see main text for details). Baseline confidence in Qwen categorical prompt steering experiment was 0.56; in Gemma it was 0.48. Baseline in Gemma numeric prompt experiment was 0.54. Significant effects observed at PANL and CC positions compared to control PANL+1 position; critically effects at the CC position occurred at *later layers* compared to those at PANL. Comparable effects observed in both models except for activation steering where stronger effects observed in Gemma.

```
You will be shown a question, together with a response you previously provided
before.
Question:  What is the name for a hare's nest?
**Your response**:  A hare is a small, fast-running animal, and its nest is
typically a simple depression in the ground, often lined with grass and fur.  [...
reasoning trace truncated ...]  However, the specific name for a hare's nest is
``form.''
**Answer**:  form
Classify your confidence into one of the following classes based on how likely
the answer above is to be correct (NO REASONING OR EXPLANATION):

- ``No chance'' (0.0--0.1)
- ``Really unlikely'' (0.1--0.2)
  [...  classes omitted ...]
- ``Almost certain'' (0.9--1.0)

Each category reflects the probability that the answer is correct.
At the very end of your output, format your confidence as
**Confidence**:  $CLASS
where CLASS is one of the names (only the names without the probability ranges)
of the classes above.
**Confidence**:
```

*Figure 23.* **Example prompt and output for Magistral Small 24B on TriviaQA**. The model's full reasoning trace (truncated here for space; max_tokens = 1024) and extracted answer are presented back to the model, which then classifies its confidence. The model's response thus consists of its *reasoning trace* followed by its *final answer*.

```
Answer the following question.  Think step by step and give your final answer.
Question:  What is the name for a hare's nest?
After your reasoning, state your final answer as:
**Answer**:  $ANSWER
where ANSWER is your final answer stated as concisely as possible.
```

*Figure 24.* **Phase 1 prompt for Magistral Small 24B on TriviaQA**. The model is instructed to reason step-by-step before producing its final answer (max_tokens = 1024). The resulting reasoning trace and extracted answer are then provided back to the model in Phase 2 (Figure 23) for verbal confidence elicitation.

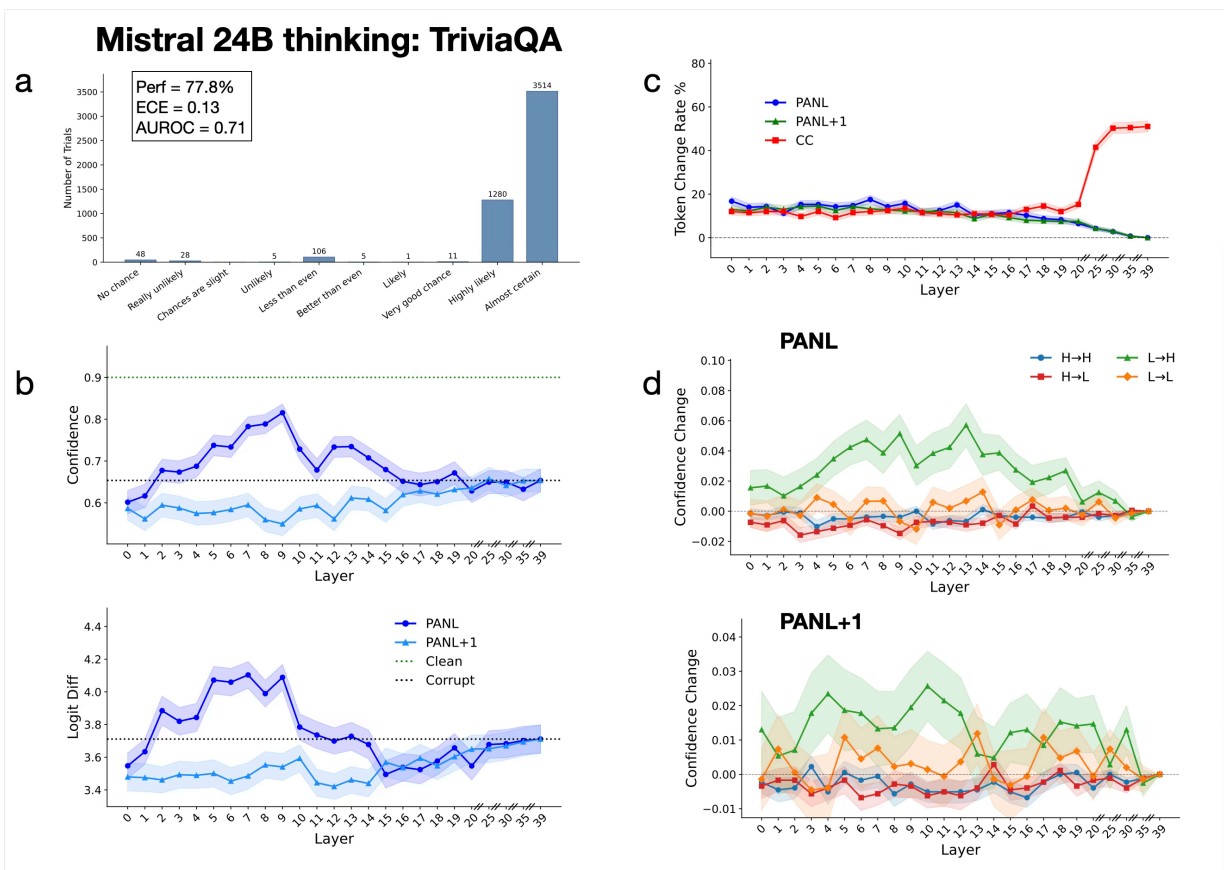

*Figure 25.* **Results of Mistral 24B reasoning model ("Magistral small 2506") on TriviaQA dataset.** a) **Distribution of confidence ratings.** Note the model outputs primarily high confidence responses but is reasonably well calibrated. b) **Activation patching.** Upper panel – change in confidence from clean baseline. PANL and control position (PANL+1) shown. lower panel – change in logit difference (i.e. between clean argmax token and mean of alternative classes) from clean baseline. c) **Activation noising.** Change in first token outputted by model (i.e. index of disruption). No effect of noising at PANL (compared to the control PANL+1 position) were observed – possibly due to information about verbal confidence being distributed across the reasoning trace, peaking at the very last token of the trace, before the final answer (see Figure 26). Note: confidence colon (CC; the very last token of the prompt – see Figure 23) is shown where significant effects were observed due to noising. d) **Activation swap.** PANL (upper panel) and control position (PANL+1; lower panel) shown. As with Gemma 3 27B (MMLU), asymmetric effects: L→H (low confidence recipient trial receives high confidence donation) direction dominating – possibly due to the tendency of the model's confidence ratings to be dominated by high confidence responses (shown in panel a)

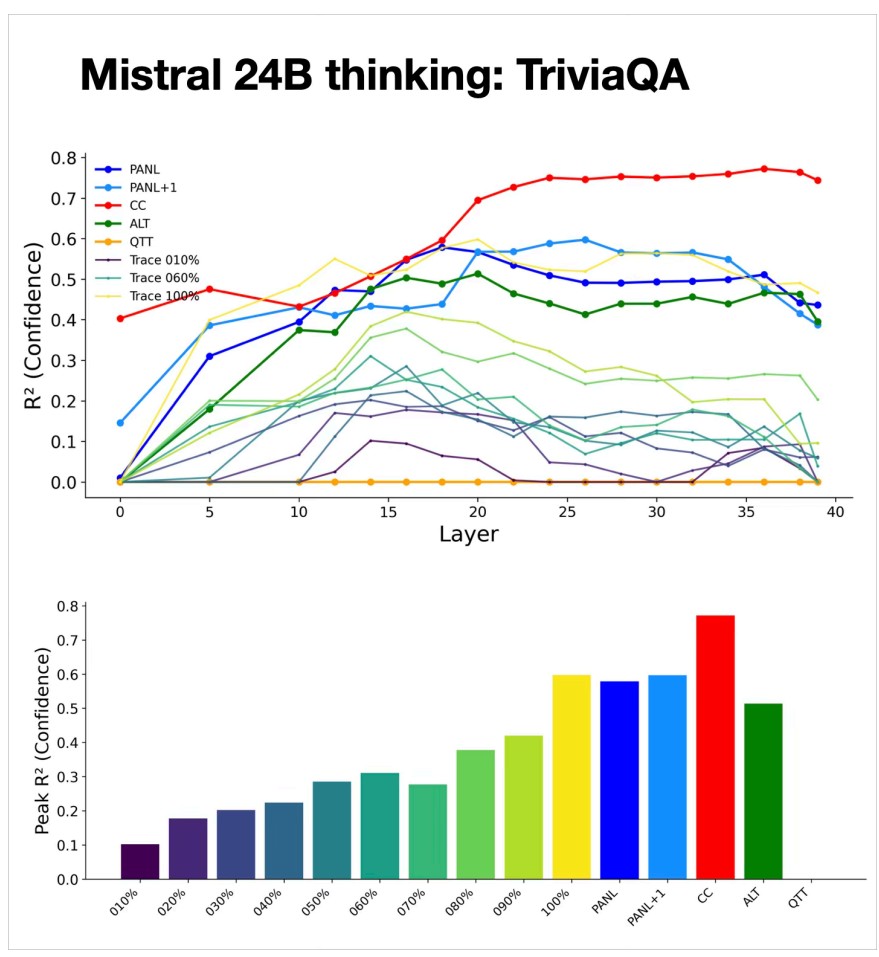

*Figure 26.* **Results of decoding in Mistral 24B reasoning model ("Magistral small 2506")**. As in the main experiments, linear probes were trained on residual stream activations (40 layers) at PANL, PANL+1, confidence colon (CC; the very last token in the whole prompt after **Confidence** – see Figure 23), last token of final answer (ALT), intermediate tokens of reasoning trace (from a token at the beginning to the end of the trace in 10% increments; hence "Trace 100%" is the very last token of the trace (usually a period)), and a control position (third question token (QTT)). We report the variance that activations at a given token explain in verbal confidence ($R^2$). Upper panel: $R^2$ for each token, across the 40 layers of the model. Lower panel: magnitude of peak $R^2$ for each token(i.e. across all layers). Note how information relating to verbal confidence increases across the reasoning trace, peaking at the very last token, where information is comparable to PANL.

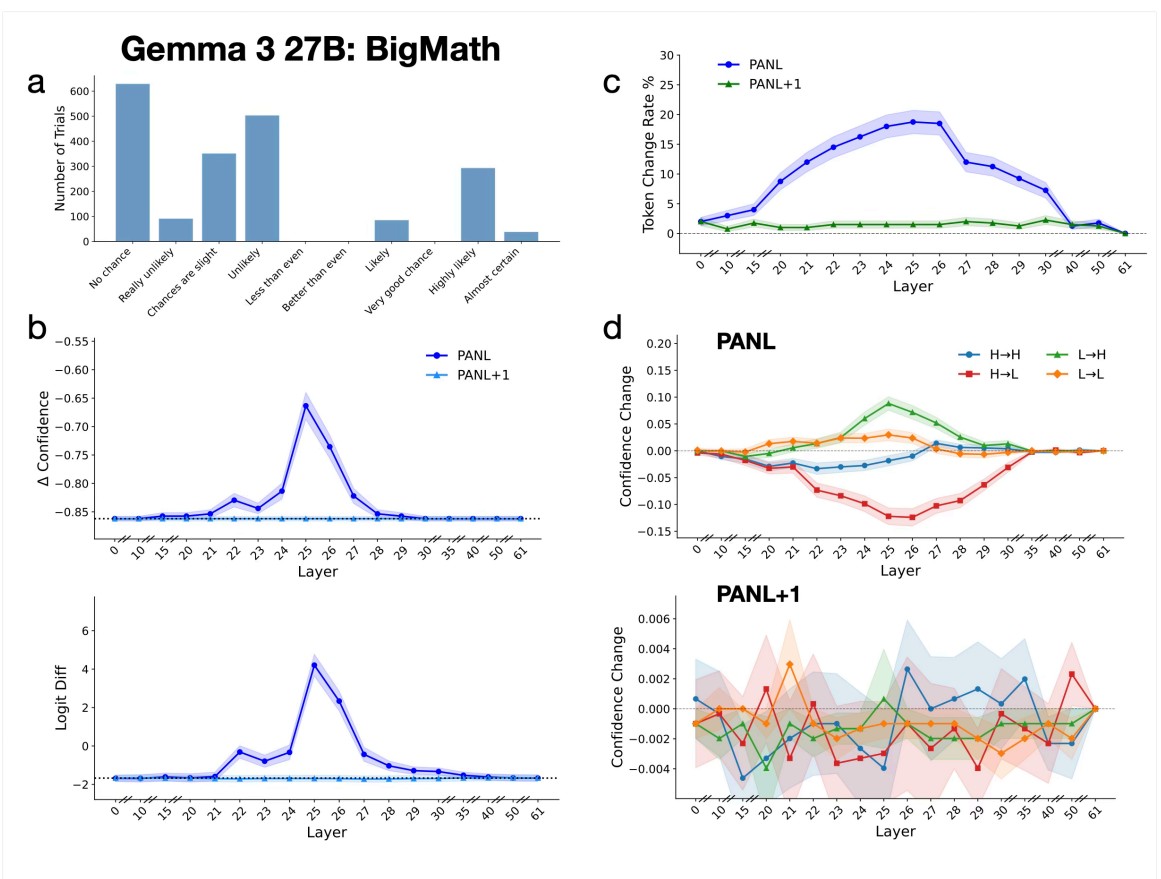

*Figure 27.* **Results of Gemma 3 27B on BigMath dataset** (Albalak et al., 2025). a) **Distribution of confidence ratings.** Overall performance 40.2% correct. b) **Activation patching.** Upper panel – change in confidence from clean baseline. PANL and control position (PANL+1) shown. lower panel – change in logit difference (i.e. between clean argmax token and mean of alternative classes) from clean baseline. c) **Activation noising.** Change in first token outputted by model (i.e. index of disruption). d) **Activation swap.** Same-confidence swaps (H→H, L→L) control for generic content-related effects due to introducing activations from a different trial; cross-confidence swaps (H→L, L→H; high confidence recipient trial receives low confidence donation, and low confidence recipient trial receives high confidence donation, respectively) isolate confidence-specific transfer. Confidence change from original (i.e. recipient trial) shown. PANL (upper panel) and control position (PANL+1; lower panel) shown.

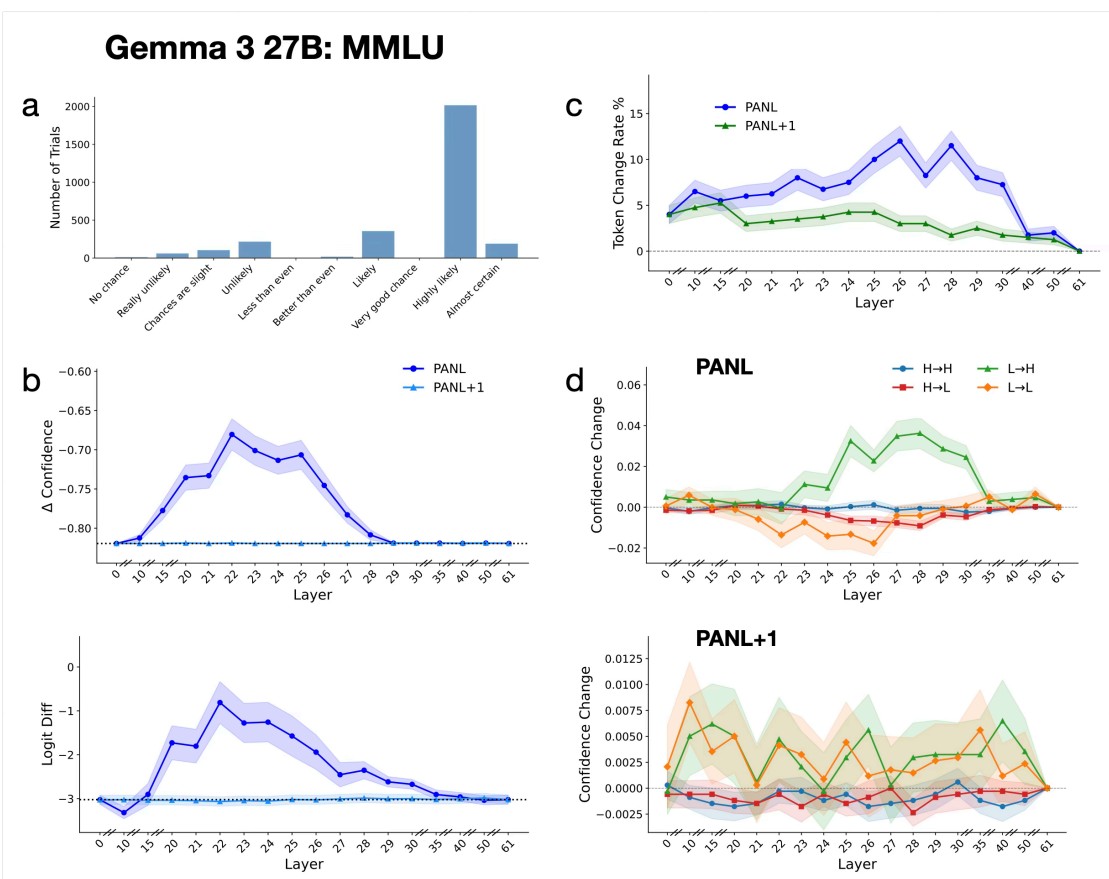

*Figure 28.* **Results of Gemma 3 27B on MMLU (multiple-choice) dataset** (Hendrycks et al., 2021). a) **Distribution of confidence ratings.** Note the model outputs primarily high confidence responses. Overall performance 76.8% correct. b) **Activation patching.** Upper panel – change in confidence from clean baseline. PANL and control position (PANL+1) shown. Lower panel – change in logit difference (i.e. between clean argmax token and mean of alternative classes) from clean baseline. c) **Activation noising.** Change in first token outputted by model (i.e. index of disruption). d) **Activation swap.** asymmetric effects with L→H (low confidence recipient trial receives high confidence donation) direction dominating – possibly due to the tendency of the model's confidence ratings to be dominated by high confidence responses (shown in panel a). Trend for effect in the L→H direction visible (but not significant). PANL (upper panel) and control position (PANL+1; lower panel) shown.

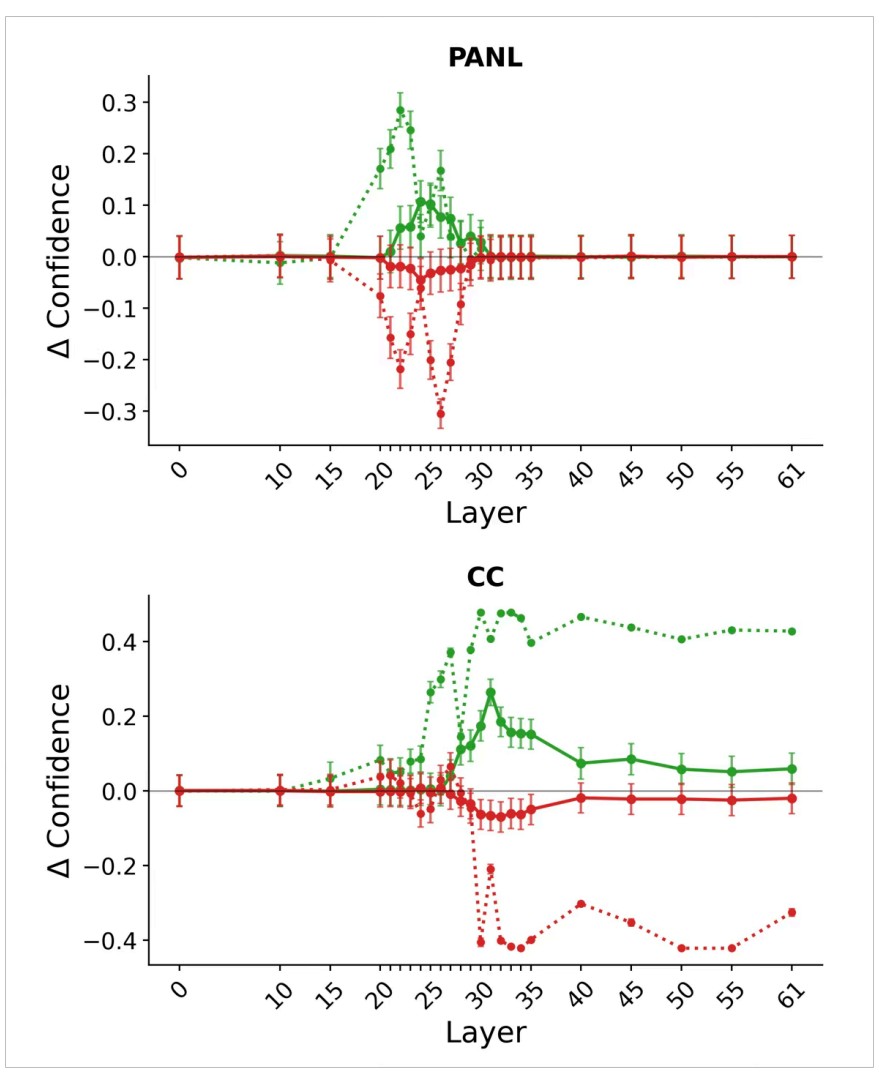

*Figure 29.* Results of Activation Steering In Gemma 3 27B at scales 1 and 10.

*Table 1.* Out-of-distribution (OOD) shift analysis for three intervention types at Layer 25 (peak effect layer). Natural distribution statistics computed from pairwise comparisons across 3000 activation collection trials. For steering, cosine similarity is between clean and steered activations. For patching, between clean-cached and corruption-propagated activations. For noising, between clean and mean-replacement activations. In all three cases, PANL and PANL+1 show comparable OOD drift, yet only PANL produces causal effects on confidence output (see Figures 2, 3, 11). Natural distribution statistics are shared across panels as they reflect properties of the model's representations at each position, independent of intervention type.

**A. Activation steering** (Layer 25)

| Position | Direction | Cosine sim | Norm ratio | Natural cos [p5, p95] | Natural NR [p5, p95] |
|---|---|---|---|---|---|
| PANL | High (s=2) | 0.999 | 0.96 | [0.998, 1.000] | [0.90, 1.10] |
| PANL | Low (s=2) | 0.999 | 1.04 | [0.998, 1.000] | [0.90, 1.10] |
| PANL | High (s=5) | 0.992 | 0.91 | [0.998, 1.000] | [0.90, 1.10] |
| PANL | Low (s=5) | 0.994 | 1.10 | [0.998, 1.000] | [0.90, 1.10] |
| PANL+1 | High (s=2) | 0.999 | 1.04 | [0.997, 1.000] | [0.90, 1.11] |
| PANL+1 | Low (s=2) | 0.998 | 0.96 | [0.997, 1.000] | [0.90, 1.11] |
| PANL+1 | High (s=5) | 0.995 | 1.10 | [0.997, 1.000] | [0.90, 1.11] |
| PANL+1 | Low (s=5) | 0.993 | 0.91 | [0.997, 1.000] | [0.90, 1.11] |

**B. Activation patching** (Layer 25)

| Position | Role | Pre-patch cos | Pre-patch NR | Natural cos [p5, p95] | Natural NR [p5, p95] |
|---|---|---|---|---|---|
| PANL | Causal | 0.999 | 0.94 | [0.998, 0.999] | [0.90, 1.10] |
| PANL+1 | Control | 0.999 | 0.98 | [0.998, 0.999] | [0.90, 1.11] |

**C. Activation noising** (Layer 25)

| Position | Role | Cosine sim | Norm ratio | Natural cos [p5, p95] | Natural NR [p5, p95] |
|---|---|---|---|---|---|
| PANL | Causal | 0.999 | 0.995 | [0.998, 1.000] | [0.90, 1.10] |
| PANL+1 | Control | 0.999 | 0.999 | [0.997, 1.000] | [0.90, 1.11] |

# C. Supplemental Methods

## C.1. Experiments with Categorical Confidence Prompt in Gemma 3 27B

### C.1.1. TECHNICAL DETAILS

For categorical confidence experiments (Gemma 3 27B and Qwen 2.5 7B), we performed single forward passes and extracted next-token logits, as categorical confidence is determined by the first generated token. For both models, we verified that forward-pass argmax tokens matched generation outputs and corresponded to valid confidence classes. Critically, the first token of each confidence class was unique (see Figure 8). Gemma 3 27B (`google/gemma-3-27b-it`) was loaded via Hugging Face. For attention analysis experiments requiring access to attention weights, an eager attention implementation was used. The model has 62 layers; layer sweeps were conducted across layers 0–61 with dense sampling in layers 20–35 where effects were strongest. The model was run in evaluation mode with greedy decoding (temperature = 0).

### C.1.2. GENERATION OF ANSWERS FOR USE IN CATEGORICAL PROMPT

In phase 0, Gemma was presented with a prompt essentially identical to that shown in Figure 8, except that confidence instructions appeared at the start. This phase was used to generate the model's answers for use in the main confidence rating experiment. Duplicates were removed from the TriviaQA dataset downloaded via the HuggingFace platform.

### C.1.3. ACTIVATION STEERING

We collected residual stream activations at several critical and control points in the prompt across layers in the network, using a separate set of questions as used in the main activation steering experiment (n = 3000). The critical points of interest were the post-answer-newline token and the confidence-colon token (see Figure 8). The control positions were the first-confidence-colon token (i.e. the token preceding the appearance of the phrase "$CLASS", and following the instruction about confidence classes); the post-answer-newline-plus1 token (i.e the first token after the PANL token). We also examined the effect of steering at the first and last tokens of the answer. For steering and other experiments we sampled layers 0, 10, 15, 20–35 (densely sampled), 40, 50, and 61.

We created high and low confidence vectors by contrasting high and low confidence trials (all trials that the model scored correctly), following standard procedures in activation steering (Turner et al., 2023; Stolfo et al., 2024a; Panickssery et al., 2023; Hua et al., 2025).

**Creation of high- and low-confidence steering vectors:** We constructed steering vectors by contrasting the 25 highest ranked trials (i.e. "Almost certain") by confidence and 25 lowest ranked trials (i.e. "No chance").

$\mathcal{H} = 25$ high-confidence trials, $\qquad \mathcal{L} = 25$ low-confidence trials.

$$\mathbf{v}_{\text{high}} = \mu(\mathcal{H}) - \mu(\mathcal{L}), \qquad \mathbf{v}_{\text{low}} = -\mathbf{v}_{\text{high}},$$

where $\mu(\cdot)$ denotes the mean residual stream activity across the selected trials.

A high-confidence steering vector was created by subtracting the mean of the low confidence trial activity vectors from the mean of the high confidence trial activity vectors. Steering vectors were scaled to 3% of the residual norm at each layer and multiplied by a constant of either 2 or 5. These values were chosen empirically, and do not induce out-of-distribution effects (see D.4). We show that scales of 1 and 10 also have effects in Figure 29. The low-confidence vector was defined as the inverse of the high-confidence vector. At test time, residual stream activity in the network at a given layer was additively modulated as:

$$\tilde{\mathbf{r}}^{(l)} = \mathbf{r}^{(l)} + \alpha\,\mathbf{v}^{(l)},$$

$\mathbf{r}^{(l)}$ : residual stream activations at layer $l$,

$\mathbf{v}^{(l)}$ : steering vector at layer $l$ (scaled to 3% of residual norm),

$\alpha$ : steering strength constant, $\alpha \in \{2, 5\}$

There were 200 questions per layer per position used for steering. To ensure a balanced selection of trials – given that the model tended to pick higher confidence classes (see Figure 9) – 1/2 of these trials were randomly sampled from the top 3 confidence classes and 1/2 from the bottom 3 classes.

### C.1.4. ACTIVATION PATCHING

**Corruption of Answer Tokens via Mean Ablation** To test whether specific position-layer combinations are sufficient for confidence computation, we use a corrupt-and-restore procedure following Meng et al. (2022); Heimersheim & Nanda (2024); Zhang & Nanda (2023); Wang et al. (2023). We first disrupted the model's access to answer information through mean ablation of answer tokens. Let $\mathbf{x}_i^{(0)}$ denote the embedding of token $i$ in the input sequence, and let $\mathcal{A} = \{a_1, \ldots, a_k\}$ denote the set of answer token positions. We computed mean activations from a calibration set $\mathcal{C}$ of 100 trials (50 high-confidence, 50 low-confidence, disjoint from the test set):

$$\bar{\mathbf{x}}_j^{(0)} = \frac{1}{|\mathcal{C}|} \sum_{c \in \mathcal{C}} \mathbf{x}_{j,c}^{(0)} \qquad (1)$$

where $\mathbf{x}_{j,c}^{(0)}$ is the embedding at answer position $j$ for calibration trial $c$. For each test trial, we replaced all answer token embeddings with their corresponding mean activations, propagating this corruption through the entire forward pass.

**Patching Procedure**  Let $\mathbf{h}_p^{(\ell)}$ denote the residual stream activation at position $p$ after layer $\ell$. For a given test trial, let $\mathbf{h}_p^{(\ell,\text{clean})}$ denote the activation from a clean forward pass (no corruption) and $\mathbf{h}_p^{(\ell,\text{corrupt})}$ denote the activation when answer tokens have been mean-ablated. Our patching intervention selectively restores the clean activation at position $p$ and layer $\ell$:

$$\mathbf{h}_p^{(\ell,\text{patched})} = \mathbf{h}_p^{(\ell,\text{clean})} \qquad (2)$$

while all other positions retain their corrupted activations. This intervention was applied after the MLP block at each layer. We tested three positions: PANL (post-answer newline), CC (confidence colon), and PANL+1 (control position). We restricted analysis to high-confidence trials because these provide the clearest test: corruption should substantially reduce confidence, whereas low-confidence trials are already near floor and thus less sensitive to disruption. There were 200 high confidence (top three classes) selected based on the original Phase 0.

### C.1.5. METRICS USED IN PATCHING AND OTHER EXPERIMENTS

**Logit Difference.**  As a generalization of (Wang et al., 2023), we define logit difference as the logit of the original confidence class minus the mean logit of alternative confidence classes:

$$\Delta_{\text{logit}} = z_{y^*} - \frac{1}{K-1} \sum_{k \neq y^*} z_k \qquad (3)$$

where $z_{y^*}$ is the logit of the clean trial's confidence class, $z_k$ are logits of the $K = 10$ confidence classes, and the sum is over the 9 alternative classes. This metric captures how strongly the model favors its original prediction relative to alternatives.

**Confidence.**  We define confidence as the midpoint of the predicted confidence class's probability range. For example, "Highly likely" (0.8–0.9) corresponds to a confidence of 0.85.

**First Token Change Rate.**  We measure the proportion of trials where the argmax token after intervention differs from the clean baseline:

$$\text{Change Rate} = \frac{1}{N} \sum_{i=1}^{N} \mathbb{1}\!\left[ \underset{k}{\arg\max}\, z_k^{(\text{patched},i)} \neq \underset{k}{\arg\max}\, z_k^{(\text{clean},i)} \right] \qquad (4)$$

**Recovery**  To quantify how effectively patching restores model behavior, we compute percent recovery for each metric $M$:

$$\text{Recovery}_M = \frac{M_{\text{patched}} - M_{\text{corrupt}}}{M_{\text{clean}} - M_{\text{corrupt}}} \times 100\% \qquad (5)$$

A recovery of 100% indicates complete restoration of clean behavior; 0% indicates no improvement over the corrupted baseline. For first token change rate, where lower values indicate recovery (clean baseline has 0% change), we invert the metric:

$$\text{Recovery}_{\text{token}} = \frac{\text{Rate}_{\text{corrupt}} - \text{Rate}_{\text{patched}}}{\text{Rate}_{\text{corrupt}}} \times 100\% \qquad (6)$$

### C.1.6. ACTIVATION NOISING EXPERIMENT

To test whether representations at PANL and CC are *necessary* for confidence reporting, we performed activation noising experiments using mean ablation. For each position of interest, we replaced its residual stream activation with the mean activation computed from a balanced calibration set of 100 trials (50 high-confidence trials from the top three classes: "Highly likely," "Very good chance," "Almost certain"; and 50 low-confidence trials from the bottom three classes: "No chance," "Really unlikely," "Chances are slight"). This calibration set was disjoint from the test set.

Notably, mean ablation does not push the model toward a semantically meaningful "neutral" confidence state. The mean of activations encoding high and low confidence is not itself an encoding of medium confidence—analogous to how averaging word embeddings for "brilliant" and "terrible" does not yield a representation of "mediocre." Rather, mean ablation disrupts the position's contribution to the computation by replacing trial-specific information with an uninformative average.

We tested three positions (n=400 trials per layer per position): PANL (post-answer newline), CC (confidence colon), and PANL+1 (control position). For each position, we applied mean ablation at individual layers across the network and measured the resulting disruption to the model's confidence output.

We assessed disruption using two complementary metrics of confidence reporting disruption: logit difference change and first token change rate. If a position is necessary for

confidence computation, ablating it should reduce logit difference and increase the first token change rate. If ablation has no effect, the position is not necessary for the model's confidence-reporting mechanism.

### C.1.7. ACTIVATION SWAP EXPERIMENT

We implemented activation swapping in a $2 \times 2$ factorial design crossing *recipient confidence* (high vs. low) with *donor confidence* (high vs. low), yielding four conditions: High→High, High→Low, Low→High, and Low→Low. The same-confidence conditions (High→High and Low→Low) control for generic cross-trial substitution effects—any swap introduces a foreign internal state that may disrupt processing independent of confidence through content-related effects. By comparing cross-confidence swaps to same-confidence swaps on the *same recipient trials*, we isolate effects attributable specifically to the donor's confidence level.

We fixed the set of recipient trials within each regime: the same 400 high-confidence recipients were used in both High→High and High→Low conditions, and the same 400 low-confidence recipients in both Low→Low and Low→High conditions. Trials were partitioned by the model's clean confidence report into high-confidence ("Highly likely", "Very good chance", "Almost certain"; $N > 400$ available) and low-confidence ("No chance", "Really unlikely", "Chances are slight"; $N = 221$ available, sampled with replacement) sets.

To control for prompt-length effects on attention patterns and decoding dynamics, we matched donor trials to recipient trials on tokenized question and answer length using quantile bins. Matching was highly successful: question-length bins matched in 100% of cases, and answer-length bins in 94–100% of cases (mean $|\Delta L_Q| \approx 1.5$–2.7 tokens; mean $|\Delta L_A| \approx 0.3$–0.5 tokens). We therefore interpret differences between cross-regime and within-regime swaps as reflecting confidence-specific information carried by PANL, rather than generic length-mismatch disruption.

### C.1.8. DECODING CONFIDENCE INFORMATION

To assess where confidence-relevant information is represented across the network, we trained linear probes on residual stream activations at each layer and token position. We extracted activations from 3,000 TriviaQA trials at positions of interest including PANL, PANL+1, and CC. Additionally, we probed other tokens in the prompt (e.g., intermediate tokens that corresponded to the first token of each confidence class as listed in the confidence reporting instruction part of the prompt) to determine how widespread the encoding of confidence information is across the model. For each position-layer combination, we trained two types of probes using 5-fold cross-validation. For predicting binary

correctness, we fit L2-regularized logistic regression and report AUROC. For predicting verbal confidence (treated as a continuous variable using class midpoints), we fit Ridge regression and report $R^2$. All activations were $z$-scored prior to fitting.

### C.1.9. VARIANCE PARTITIONING USING LOG-PROBABILITY BASELINES

**Token log-probability extraction.** To obtain a white-box measure of model confidence, we extracted token-level log-probabilities during Phase 0 generation. For each generated sequence, we computed the log-probability of each token $t_i$ given the preceding context, $\log p(t_i \mid t_{<i}, \mathbf{x})$, where $\mathbf{x}$ denotes the input prompt. From these we computed six answer log-probability summaries, each restricted to the answer span (identified by mapping the extracted answer string back to the generated token sequence): the length-normalized mean

$$\bar{\ell} = \frac{1}{n} \sum_{i=1}^{n} \log p(t_i \mid t_{<i}, \mathbf{x}), \qquad (7)$$

the minimum, maximum, and variance of per-token log-probabilities across the $n$ answer tokens, and the log-probabilities of the first and last answer tokens individually. Length-normalization of the mean controls for variation in answer length, ensuring that longer answers are not penalized simply for having more tokens.

**Simple correlations.** As a scalar diagnostic, we computed correlations between length-normalized answer logprobs and verbal confidence ratings from both Phase 0 (same run) and Phase 1 (different run with identical questions but answers provided in the prompt). Logprobs explained only 4.9% of variance in within-run verbal confidence ($r = 0.23$, $R^2_{\mathrm{CV}} = 0.049$) and 8.4% in cross-run verbal confidence ($r = 0.29$, $R^2_{\mathrm{CV}} = 0.084$). The model's verbal confidence showed moderate consistency across runs, with Phase 0 and Phase 1 ratings correlating at $r = 0.63$ ($R^2_{\mathrm{CV}} = 0.40$), indicating that confidence judgments are relatively stable across repeated presentations of the same question-answer pair.

**Variance partitioning procedure.** To test whether residual stream activations explain confidence variance *beyond* what is captured by answer log-probabilities, we partitioned variance between the two feature sets. For each (layer, position) combination, we fit three cross-validated Ridge regressions ($K = 5$ folds) predicting verbal confidence midpoint:

- $R^2_{\mathrm{act}}$: activations alone (residual stream at that layer/position),

- $R^2_{\mathrm{base}}$: log-probability baselines alone (either one of the six summaries, or all six concatenated),

- $R^2_{\text{both}}$: activations and baselines concatenated.

The variance uniquely attributable to activations is then $R^2_{\text{unique}} = \max(0, R^2_{\text{both}} - R^2_{\text{base}})$ — the gain in predictive power from adding activations to a regression that already includes the log-probability baselines. All features were z-scored prior to fitting; Ridge regularization $\alpha = 1.0$. We ran this partitioning twice: against each of the six baselines individually (coloured lines in Figure 14), and against all six combined into a single regression (black line). The combined-baseline analysis is the most conservative test, since it credits the baselines with any redundant linear information across the six summaries.

### C.1.10. ATTENTION BLOCKING METHOD

To trace information flow during confidence generation, we employ attention knockout (Geva et al., 2023), which blocks attention edges between specific token positions. For a source position $s$ and target position $t$, we set the attention weights $\alpha_{t \to s}$ to zero across all attention heads within a specified layer range, preventing information from flowing from $s$ to $t$. We measure the effect on confidence output using two metrics: (1) first token change rate—the proportion of trials where the argmax confidence token differs from the unmodified baseline; and (2) logit difference change—the shift in the margin between the baseline token's logit and the mean logit of alternatives.

We use a minimal prompt format that elicits numeric confidence (0–9) and reduces the number of intermediate tokens between PANL and CC (see Figure 16). This minimal prompt was used to test three key pathways: (1) CC→Q+A, to test whether CC computes confidence via just-in-time attention to question and answer tokens; (2) CC→PANL, to test whether CC retrieves cached confidence information from PANL; and (3) PANL→A and PANL→last_A, to trace how PANL gathers confidence information from answer tokens. The first and third findings were confirmed in complementary experiments using the full categorical prompt (see §D for details), where blocking all downstream tokens from attending to answer positions replicated the disruption observed with the minimal prompt.

Following Geva et al. (2023), we note that this method does not account for information that may have passed between positions at earlier layers prior to the blocking intervention. Thus, null results should be interpreted cautiously, while positive results provide evidence that the blocked pathway carries task-relevant information at the targeted layers.

### C.2. Experiment using Numeric Confidence Prompt in Gemma 3 27B

We used a prompt derived from (Devic et al., 2025; Mei et al., 2025) (see Figure 15). For numeric confidence ex-periments (Gemma 3 27B), we used max-new-tokens=4 to generate the full confidence value (e.g., "95"). The positions of the PANL, PANL+1, FCC (first confidence colon) and CC tokens were analogous to their positions in the categorical confidence prompt.

**Metrics for numeric confidence** Analogous to the categorical confidence prompt experiment, we track three metrics in the numeric confidence prompt experiments. *Confidence change* is the difference between the integer value (0–100) generated in the baseline condition and the (e.g.) noised condition; *First digit change rate* measures the proportion of trials where the first generated digit differs from the clean baseline—analogous to the first token change rate used for categorical confidence; *Logit difference* is computed as the logit of the clean trial's first digit minus the mean logit of the other nine digits (0–9), capturing how strongly the model favors its original confidence judgment over alternatives.

**Multi-token numeric confidence and KV-cache considerations** Numeric confidence involves multi-token generation (e.g., "9→5"). All interventions occur during prefill, modifying cached key–value representations that subsequent digits attend to. The second digit is generated by attending to both the cached prompt and the first digit, meaning effects on later digits reflect both the original intervention and changes propagated through earlier tokens.

### C.3. Experiment using Qwen 2.5 7B

Qwen 2.5 7B Instruct was tested on the categorical prompt (see Figure 8). The model was loaded via Hugging Face. Qwen showed reasonable calibration (ECE = 0.06, AUROC = 0.65)(see Figure 18).The model has 28 layers; we used a dense layer sweep across layers 0–27. We used the same procedures as described for Gemma. The model was run in evaluation mode with greedy decoding (temperature = 0). We used the same TriviaQA subset as for Gemma, with 3000 trials for activation collection. Due to the distribution of confidence ratings (see Figure 18) low confidence trials were sampled from the "Unlikely" class, and high confidence trials from the "Likely" class. 150 questions per condition per layer were used for steering; 200 questions for activation patching; 300 questions for noising (200 high confidence, 100 low confidence trials); 200 questions per condition for the activation swap experiment.

### C.4. Experiments using Magistral Small 24B Reasoning

We used Magistral Small 2506, a 24B-parameter 40 layers dense decoder-only transformer released by Mistral AI. We loaded the model from the HuggingFace checkpoint `mistralai/Magistral-Small-2506`, with `attn_implementation="eager"` to enable attention

weight access for the attention blocking experiments. Max-tokens was 1024, greedy decoding was used.

We ran the initial behavioral experiment (i.e. answer and confidence collection) on 5000 TriviaQA questions ($N = 4998$ after filtering for valid answers and confidence classifications). For the activation experiments, we sampled a stratified subset of 3000 trials, retaining all trials with verbal confidence $\leq 0.7$ and filling the remainder by random sampling from high-confidence trials, to preserve the limited pool of low-confidence trials needed for H/L contrasts.

**Activation Patching, Noising, and Swap** We performed activation patching, noising, and swap experiments on Magistral Small 24B in the context of the Trivia QA dataset. The methodological framework matched that used for Gemma 3 27B; below we describe only the Magistral-specific adaptations.

**Common adaptations.** All three experiments used the CoT prompt versions (Figure 23, Figure 24). In the verbal confidence prompt the full reasoning trace and extracted answer appear within the response block preceding the PANL newline. We defined PANL as the newline token terminating the response block, and PANL+1 as the subsequent token. Trial selection drew from the extremes of the confidence distribution (sorted by midpoint) rather than uniformly within each high/low band, providing the cleanest contrast given the model's ~92% concentration in "Almost certain". The logit difference metric tracked the clean trial's predicted confidence-class token across all intervention conditions (fixed target), rather than the argmax of each condition.

**Activation Patching.** We used the corrupt-then-restore paradigm as in the main paper. The key adaptation concerns the corruption scope: in the non-CoT setting, confidence-relevant content is concentrated in the short answer string, but in CoT it is distributed across the full reasoning trace. We therefore corrupted the embeddings of all tokens spanning the question and the entire response block (reasoning trace plus final answer string), excluding the PANL newline itself and all downstream classification tokens. Unlike the main Gemma experiments, where trials were drawn from top-three/bottom-three classes without further ordering, we additionally sorted by confidence midpoint and selected the most extreme exemplars within each band. This was motivated by Magistral's stronger concentration of probability mass in the highest class (high % of "Almost certain" response).

**Activation Noising.** Mean ablation followed the Gemma protocol. Mean activations were collected at PANL and PANL+1 from a 100-trial calibration set (50 highest + 50 lowest confidence). The test set comprised $n = 400$ trials

selected from the extremes of the confidence distribution (200 lowest-confidence + 200 highest-confidence), shuffled. We tested CC as a control position (rather than as a primary position) in a separate analysis given its near-certain ceiling.

**Activation Swap.** The $2 \times 2$ factorial design (recipient confidence $\times$ donor confidence) matched the Gemma experiments. The recipient pool comprised 400 high- and 400 low-confidence sources, selected from the extremes. Donors were length-matched to recipients on both question length and reasoning trace length using 10-quantile bins on each axis. Swaps were performed at PANL and PANL+1.

**Activation collection and decoding analyses.** For each trial in the stratified set, we collected activations at 10 trace positions (evenly spaced at 10% increments across the whole reasoning trace, from the trace start to the final answer token within the response block) and at five standard positions: post-answer-newline (PANL), PANL+1, confidence-colon (CC), the last answer token, and the third question token (control). Probing methodology followed the main experiments: e.g. ridge regression for verbal confidence (5-fold cross-validated $R^2$).

## D. Supplemental Results

### D.1. Attention Blocking with Categorical Confidence Prompt

To complement the attention blocking experiments using the minimal numeric prompt (main text), we conducted parallel experiments using the full categorical confidence prompt (Figure 8). These experiments provide convergent evidence against the just-in-time hypothesis; however, they do not reveal the direct pathway from PANL to CC, likely because confidence information can flow through multiple intermediate template tokens, masking the effect of single-edge (CC→PANL) blocking.

**CC does not compute confidence from scratch.** Blocking CC's attention to question tokens, answer tokens, or both produced minimal effects (~10% change rate), equivalent to a control condition blocking CC→PANL+1 (Figure 21). This rules out the just-in-time hypothesis.

**No direct CC→PANL pathway detected.** Unlike the minimal prompt results reported in the main text, blocking CC→PANL produced no effect with the categorical prompt. This null result likely reflects redundant information routing: confidence information cached at PANL can reach CC through multiple intermediate template tokens, so blocking the direct pathway alone is insufficient to disrupt confidence output.

**PANL caches confidence information.** Blocking all later tokens from attending to PANL (ALL→PANL) produced a

20% change rate and substantial logit difference reduction, confirming that PANL serves as a confidence cache even when direct CC→PANL blocking is ineffective.

**Answer tokens are the primary information source.** Blocking all attention to the last answer token (ALL→last_A) produced a 50% change rate (Figure 21). Critically, preserving only the PANL→last_A pathway reduced this effect to 20%, demonstrating that PANL reads confidence-relevant information from answer tokens and relays it downstream. The same pattern held for all answer tokens: ALL→A produced 70% change, reduced to 40% when preserving PANL→A.

These results converge with findings from the minimal prompt to support the cached retrieval model: confidence information originates at answer tokens, is read by PANL, and flows to CC for verbalization.

## D.2. Generalization to a Reasoning Model: Magistral Small 24B

To test whether the cached retrieval hypothesis generalizes to reasoning models — where a chain-of-thought trace intervenes between the question and the final answer — we replicated our main experiments on Magistral Small 2506, a 24B-parameter open-weight reasoning model (40 layers) from Mistral AI. We used the TriviaQA dataset as in the main experiments. Methodological details specific to Magistral, including the two-phase prompt design, trial stratification, and activation collection across reasoning trace positions, are provided in §C.4. Magistral was reasonably well calibrated and showed a similar concentration of probability mass in high-confidence classes as Gemma (Figure 25a).

**Activation patching.** Patching of PANL representations produced substantial recovery of confidence following corruption of the question and full response block (reasoning trace plus final answer), while patching at the control PANL+1 position had no effect (Figure 25b). This replicates the central patching result from Gemma — confidence-relevant information is sufficient at PANL to drive verbal confidence output, but not at the immediately adjacent control position.

**Activation noising.** Mean ablation of PANL produced no effect on verbal confidence beyond the PANL+1 control, in contrast to Gemma where PANL noising disrupted confidence reporting (Figure 25c). However, noising at the confidence-colon (CC) position produced substantial disruption. This dissociation between patching (PANL effective) and noising (PANL ineffective) is illuminated by the decoding results: in the CoT regime, confidence-relevant information is distributed across the reasoning trace especially toward its end (Figure 26). This distributed encoding

means that ablating PANL alone leaves intact a redundant trace-distributed signal that CC can retrieve directly.

**Activation swap.** Cross-confidence swaps at PANL (L→H and H→L) produced systematic directional shifts in confidence beyond same-confidence controls, with no effect at PANL+1 (Figure 25d). As observed in Gemma on MMLU, the L→H direction dominated — plausibly reflecting Magistral's strong concentration of probability mass in high-confidence classes, which creates a near-ceiling for high-confidence trials and leaves the low-to-high direction as the most informative test. The swap result confirms that PANL representations carry confidence-specific information that transfers across trials.

**Decoding.** Linear probes trained on residual stream activations revealed that confidence information is decodable from PANL ($R^2$ at peak layer comparable to that of the final trace token), from CC at later layers, and increasingly across the reasoning trace, peaking at the final trace token immediately preceding the answer (Figure 26). The control position (QTT, third question token) explained negligible variance.

**Summary.** Across patching, swap, and decoding, the Magistral results support the cached retrieval hypothesis: confidence information is encoded at the post-answer-newline position following the model's final answer, even when that answer is preceded by an extended reasoning trace.

## D.3. Gemma 3: Variance partitioning using log-probability baseline

The variance partitioning analysis reported in the main paper used a single log-probability baseline: the length-normalized mean log-probability across answer tokens, which explained $R^2_{\text{CV}} = 0.084$ of variance in verbal confidence. To rule out the possibility that other summaries of answer log-probabilities — capturing aspects such as the lowest-confidence token, the highest-confidence token, position-specific values, or token-level variability — might account for more of the verbal confidence signal, we extended the analysis to six log-probability baselines: the mean, minimum, maximum, and variance of per-token log-probabilities across the answer tokens, and the log-probabilities of the first and last answer tokens individually. Individual baselines explained between 2.5% and 10.1% of confidence variance ($R^2_{\text{CV}}$: min = 0.101, mean = 0.084, first token = 0.070, variance = 0.051, last token = 0.039, max = 0.025); combined into a single regression, all six baselines together explained only $R^2_{\text{CV}} = 0.100$, indicating that the baselines carry substantially overlapping linear information about verbal confidence. Critically, PANL activations ex-

plained $R^2_{\text{unique}} = 0.38$ at the peak layer (layer 40) *beyond* the combined six-baseline set (Figure 14, black line) — more than three times larger than the variance explained by any individual log-probability summary. This confirms that the confidence representation at PANL is not reducible to token-level probability signals, whether measured as mean, extremes, variability, or position-specific values.

### D.4. Gemma 3: Causal Interventions Remain Within the Natural Activation Distribution

A general concern for activation-based causal interventions is that perturbations may push the residual stream into out-of-distribution (OOD) regions, producing effects that reflect generic model disruption rather than the manipulation of meaningful representations. To address this, we quantified the OOD shifts induced by each of our three intervention types — activation steering, patching, and noising — and compared them against the natural pairwise variability of activations at the same positions (computed across 3000 trials in the activation collection set). We report two metrics: cosine similarity to the unperturbed activation, and norm ratio between perturbed and unperturbed activations (Table 1).

For all three intervention types, cosine similarity to clean activations exceeded 0.99 and norm ratios remained within 0.91–1.10, both well within the natural pairwise distribution observed across trials (5th–95th percentile cosine: [0.997, 1.000]; norm ratio: [0.90, 1.11]). Critically, the OOD drift induced at PANL is comparable to that induced at the control position PANL+1 across all three interventions, despite the markedly different causal effects on verbal confidence at these two positions. For example, in the patching experiment at Layer 25, PANL and PANL+1 showed nearly identical pre-patch cosine similarity to clean activations (0.999 at both positions), yet patching at PANL produced 24.3% recovery of confidence while PANL+1 produced $-1.4\%$ (i.e., no recovery). This dissociation between OOD drift (comparable across positions) and causal effect (specific to PANL) rules out the interpretation that our results are driven by generic out-of-distribution artifacts. The convergence of evidence across five distinct intervention techniques — steering, patching, noising, swap, and attention blocking — each of which perturbs the residual stream in mechanistically distinct ways, further mitigates this concern.

### D.5. Limitations and Future Work

Several limitations bound the scope of our claims. First, while we showed that the cached retrieval mechanism is robust across categorical and numeric prompt formats (§2.9), our experiments fix the prompt wording within each format, and we did not test sensitivity to small paraphrases or to persona-style framing — for example, instructing

the model to respond "uncertainly" versus "confidently". Whether such framings shift confidence by modulating the PANL representation upstream or by biasing the verbalization stage at CC is a natural extension of our intervention framework. Second, our attention blocking experiments rule out just-in-time computation at CC and confirm that PANL caches confidence information retrieved by CC, but they leave open the specific routing of this retrieval in prompts with extensive intermediate template tokens. While the minimal numeric prompt revealed a direct CC→PANL pathway, the full categorical prompt showed that confidence information likely flows through multiple intermediate template tokens. Identifying the specific attention heads that mediate PANL→CC retrieval — and characterizing how information is routed through intermediate template positions in longer prompts — is an important direction for future work.

