# OpenReview forum: "How do LLMs Compute Verbal Confidence?"
_ICML.cc/2026/Conference — ICML 2026 regular_

### Official Review · Reviewer_eMyR · 2026-03-08

**Soundness:** 4
**Presentation:** 3
**Significance:** 3
**Originality:** 3
**Overall Recommendation:** 4
**Confidence:** 4

**Summary:**

This paper investigates how large language models compute verbal confidence under explicit prompting. Using Gemma 3 27B as the primary experimental model and Qwen 2.5 7B for replication , the authors combine a broad suite of mechanistic interpretability methods. These include activation steering , activation patching , mean ablation (noising) , activation swapping , attention blocking , linear probing , and variance partitioning. The study demonstrates that confidence information is formed and cached at the post-answer newline (PANL) token. This information is subsequently retrieved at the confidence-colon (CC) token for output.

**Compliance With Llm Reviewing Policy:**

Affirmed.

**Final Justification:**

I believe the rebuttal has addressed most of my concerns, and I lean to weak acceptance of the paper.

**Key Questions For Authors:**

* Are the "unique variance" results robust when utilizing richer token probability baselines (e.g., last token log-probability, entropy, margin)?


* Can you quantify the distribution shift caused by the interventions and attempt alternative patching schemes for verification?


* Can you explicitly map the multi-hop pathways (per layer and per head) within the categorical prompt?


* How sensitive is the PANL effect to variations in delimiters, templates, languages, or tokenization strategies?


* To what extent is cached confidence localized specifically at PANL, and how much does the last answer token contribute?


* Do the PANL and CC effects hold under different decoding strategies (e.g., greedy decoding vs. sampling, or temperature variations)?


* Can you identify the specific attention heads that mediate the PANL→CC retrieval, and would intervening on them alter calibration metrics?


* Is the cached retrieval mechanism generalizable to non-factual QA (e.g., math, code, or out-of-domain tasks)?

**Limitations:**

yes

**Strengths And Weaknesses:**

## Strengths

*
**Technical Novelty and Integrated Mechanistic Tools:** The paper combines multiple causal and representational methods to study verbal confidence, which is much more persuasive than single-method approaches. The PANL→CC cached retrieval hypothesis successfully extends the cache-and-retrieve concept from factual recall to the domain of uncertainty reporting. It treats verbal confidence as a mechanistically understandable internal computation, rather than merely a behavioral correlate.


*
**Consistent and Clear Causal Evidence:** Across activation steering, patching, noising, and swapping experiments, PANL effects consistently precede CC effects. This supports the temporal sequence of confidence formation. Swap experiments successfully isolate confidence-specific information from answer-content interference. Attention blocking provides robust evidence against the just-in-time computation hypothesis under a minimal prompt. The use of null control positions further strengthens the causal interpretations.


*
**Rigorous Experimental Design:** The prompt design, metrics, and choice of controls (such as ensuring each confidence class begins with a unique token ) enhance interpretability and validity. Metrics like logit difference and first token change rate are highly appropriate for mechanistic analysis.


*
**Significance and Potential Impact:** The core questions addressed—what verbal confidence represents and when it is computed—are valuable both practically and theoretically. These findings could inform improvements in calibration, user interfaces, safety workflows, and the study of LLM metacognition. This offers practical and academic value to the ICML community.


## Weaknesses

*
**Mechanistic Conclusions Overstate the Evidence:** Although multiple interventions support cached retrieval, the paper occasionally implies this is the sole mechanism, overgeneralizing the findings. Distributed or overlapping circuits, compensatory pathways, or prompt-specific implementations might still contribute to confidence generation. For instance, patching and noising yield only partial effects. This partial recovery indicates that multi-layer or parallel pathways might be concurrently involved. Therefore, a more conservative phrasing should emphasize that cached retrieval likely dominates in the studied environment, rather than generalizing it to all LLMs or prompt types.


*
**Insufficient Baselines for 'Beyond Log-Probability' Claims:** The conclusion that PANL/CC representations encode confidence beyond log-probabilities is highly interesting, but the evidence is not yet fully conclusive. The baseline comparison primarily relies on the mean answer log-probability. Stronger baselines (e.g., last token log-probability, softmax margin, entropy trajectories, token variance, min/max log-probability, or length-normalized likelihoods) might explain some of the residual variance. Currently, the claim of an independent confidence signal remains suggestive rather than rigorously verified.


*
**Incomplete Characterization of Routing Pathways:** While attention blocking experiments are clear under the simplified prompt , the multi-hop intermediate tokens in the categorical prompt obscure the CC→PANL pathway.
The paper does not explicitly detail the intermediate hop pathways layer-by-layer or head-by-head, making the mechanistic description somewhat speculative rather than entirely empirical. The observed temporal sequence might reflect prompt-specific patterns rather than a universal architectural mechanism.


*
**Interventions May Introduce Distribution Shifts:** Patching (replacing with mean activations) and steering might produce unnatural model states, potentially impacting interpretations. Although multi-method replication mitigates this risk , the causal conclusions would be more robust if the authors quantified the out-of-distribution (OOD) shifts (e.g., via activation norms, cosine angles, or residual stream statistics) and tested alternative ablation methods.


*
**Limited External Validity:** The experiments primarily focus on no-CoT, TriviaQA-style factual question answering. The study predominantly utilizes Gemma 3 27B. Even though results were replicated on Qwen 2.5 7B and with numeric prompts, broader task validation is needed to generalize these findings to mathematics, coding, multi-hop reasoning, or adversarial QA.

---

> ### Author Rebuttal · Authors · 2026-03-31
>
> We are grateful to the reviewer for their constructive comments.
>
> **W1** We agree and appreciate this nuanced point. Our results are indeed consistent with distributed or overlapping circuits contributing to confidence generation. As we note in §2.2 (lines 216–219), partial recovery under patching is expected given recent work showing model behaviors arise from many overlapping heuristics rather than single mechanisms (Lindsey et al., 2025; Ameisen et al., 2025).  We will be careful to adopt more conservative phrasing throughout in the revised manuscript, emphasizing that cached retrieval is the dominant mechanism in our studied settings rather than the sole or universal one.
>
> **W2** We thank the reviewer for this important suggestion. We extended our variance partitioning to include five token-level probability baselines: mean log-probability (R²=0.084), minimum (R²=0.101), maximum (R²=0.025), last answer token (R²=0.039), and variance (R²=0.051). Combined, all five explained R²=0.098 of variance in verbal confidence. Critically, PANL activations explained substantial unique variance beyond all five combined (R²_unique=0.380 at peak layer ~30), confirming that PANL representations are not reducible to any combination of token-level probability summaries. These results strengthen our conclusion that PANL encodes a confidence-specific representation that goes beyond what is captured by white-box measures such as token log-probabilities.
>
> **W3** We acknowledge this limitation. As the reviewer notes, the CC→PANL pathway is clearly demonstrated in the minimal prompt but obscured by multi-hop routing through intermediate template tokens in the categorical prompt (§2.6, §D.1). We were transparent about this: the minimal prompt was designed precisely to reveal this pathway when the categorical prompt's complexity made it intractable. We agree that full head-by-head characterization of intermediate routing would strengthen the mechanistic account, and will note this as important future work. However, we would argue against the concern that temporal precedence is prompt-specific: the PANL-before-CC pattern replicates across three prompt formats (categorical, numeric 0–100, minimal 0–9), two datasets (TriviaQA, now also BigMath and MMLU), and three architectures (Gemma, Qwen, Mistral - the latter a new reasoning CoT model added during revision).
>
>
> **W4** We thank the reviewer for this insightful comment. We quantified OOD shifts for patching and steering. For patching, PANL+1 experienced comparable corruption-induced drift as PANL (cosine similarity to clean: 0.999 at both positions, within the natural pairwise range), yet showed zero recovery (-1.4% vs 24.3%) — ruling out generic OOD artifacts. For steering at scale 2, cosine similarity to clean activations was 0.999 (within natural pairwise range [0.998, 1.000]) and norm ratios were 0.96–1.04 (within natural range [0.90, 1.10]). Cross-method convergence across 5 techniques with distinct perturbation characteristics further mitigates this concern.
>
> **W5** We have substantially expanded experimental coverage. We now include two additional datasets (BigMath for mathematical reasoning, MMLU for 57 subjects spanning STEM, humanities, and social sciences),  and a third model (Mistral Small 3.1 24B, a reasoning model with CoT enabled by default).. The core findings replicate across all settings. Please see our responses to Reviewer #1 UYqi (W3; datasets and Q2: CoT) and Reviewer #2 giwM (W2, architecture) for details. We agree that coding, multi-hop reasoning, and adversarial QA remain important future directions.
>
> **Q1-3** please see above responses
>
>
> **Q4** Our results replicate across three prompt formats with different delimiters and templates, suggesting robustness. Systematic investigation of delimiter tokens, languages, and tokenization strategies is an important future direction, connecting to broader questions about newline tokens as privileged information-aggregation sites in LLMs.
>
>
> **Q5** The last answer token shows effects comparable to PANL (compare Figures 2 and 10), but is confounded — it simultaneously serves as answer content and confidence locus (§2.1, lines 155–164). Our claim is not that PANL is the sole caching site, but that it is the first unconfounded position. PANL+1 shows no causal effects across any experiment, confirming spatial specificity.
>
>
> **Q6** If we understand correctly, the reviewer asks whether the mechanism holds under non-greedy decoding. We note that our findings concern forward-pass representations at PANL and CC, which are computed before any decoding strategy is applied — the decoding strategy affects which token is selected from the output distribution, not the internal representations that produce it.
>
>
> **Q7** Identifying specific attention heads mediating PANL→CC retrieval is an important next step for future work.
>
>
> **Q8** please see W5 above — BigMath (mathematics) and MMLU (diverse domains) replicate the core findings.

---

> > ### Author Rebuttal · Reviewer_eMyR · 2026-04-02
> >
> > Thank you for the rebuttal. I appreciate the additional detail on stronger log-probability baselines, OOD-shift analysis, and the effort to qualify the main claims more conservatively. These responses are helpful.
> >
> > My remaining concern is that several of the most important new claims are described only at a high level. In particular, the added results on new datasets/models, the stronger variance baselines, and the OOD analyses would be easier to assess if the rebuttal included concrete tables/figures or an anonymous supplementary link.
> >
> > I also still have follow-up questions about the mechanistic completeness of the account, especially the unresolved multi-hop / head-level routing in the categorical prompt and the extent to which the conclusions generalize across prompts and settings. Overall, my concerns are reduced, but not fully resolved.

---

> > > ### Author Response · Authors · 2026-04-03
> > >
> > > We thank the reviewer for their constructive feedback.
> > >
> > > Point 1: We have now created a pdf which contains the new datasets/models (see Figures 1-5), details about the log-probability baselines (see Figure 6 and Table 1), and details of the OOD analyses (see Table 2).
> > >
> > > The pdf is available at:
> > >
> > > https://osf.io/cs7ph/files/2xzvp?view_only=1c6227274bcd49cba75c0147c94697e0
> > >
> > > Point 2: We appreciate the reviewer’s comment about discovering the details of information flow within the categorical prompt, and will highlight this as an important avenue for future work.
> > >
> > > We will include the figures where possible in the main paper (space permitting), otherwise they will be in the Appendix.

---

### Official Review · Reviewer_ziFg · 2026-03-08

**Soundness:** 2
**Presentation:** 2
**Significance:** 3
**Originality:** 3
**Overall Recommendation:** 3
**Confidence:** 4

**Summary:**

This paper investigates how LLMs internally generate verbal confidence, posing two questions: (1) when is confidence computed, and (2) what does confidence represent. Using Gemma 3 27B and Qwen 2.5 7B on TriviaQA, the authors conduct six mechanistic analyses and conclude that confidence is generated via a cached retrieval mechanism, with representations that go beyond token log-probabilities.

**Compliance With Llm Reviewing Policy:**

Affirmed.

**Key Questions For Authors:**

The paper addresses an interesting topic, but the experiments are insufficient and the writing logic needs improvement. I recommend the authors expand to a broader range of models, narrow the research focus to a single core question, and explicitly articulate in the experimental sections how each result supports the answer to that question.

**Limitations:**

yes

**Strengths And Weaknesses:**

### Strengths:

The research question is interesting, and the cross-validation from multiple angles is a commendable approach.

The attention blocking experiments clearly demonstrate that blocking CC's attention to question and answer tokens has minimal impact on output, providing direct causal evidence.


### Weaknesses:

* All methods employed are standard tools in mechanistic interpretability. The paper reads more like a sequential application of an existing toolkit rather than a methodological contribution. For a venue like ICML, this lack of methodological novelty is a notable shortcoming. The paper may be better suited for NLP venues such as EMNLP.
* The logical connections between experiments are unclear. Each experimental subsection reads independently, lacking transitions that explain what question the previous experiment left open and how the current one addresses it. The numerous experiments and findings are presented in a scattered manner, making it difficult to summarize the core finding in a single sentence after reading the full paper. I recommend streamlining the presentation, highlighting the most critical findings, and moving supplementary experiments to the appendix.
* The answer to the second research question ("what") is insufficient. The only evidence supporting the "richer evaluation" claim is a single variance partitioning analysis, which only tells us what confidence is not (i.e., not merely log-probabilities), but entirely lacks a positive characterization of what it is.
* Experimental coverage is limited, raising concerns about generalizability. Only two models, one dataset (TriviaQA), and one type of prompt structure (answer-first, no CoT) are tested. Multiple open-source models equally support internal representation access, and datasets should also cover different task types such as reasoning.
* Key experimental details lack justification. For instance, why were scaling factors of 2 and 5 chosen for activation steering? Was any sensitivity analysis conducted? Similar design choices throughout the paper remain insufficiently motivated.
* CoT scenarios are not covered, and the discussion of this limitation is inadequate. The paper deliberately suppresses CoT, yet many real-world applications involve CoT reasoning. Whether the cached retrieval mechanism holds under CoT conditions is an important question that needs to be addressed.
* Related work citations are incomplete. I recommend the authors cite and discuss two highly relevant works: (1) Yang et al. (2024) "On Verbalized Confidence Scores for LLMs"; (2) Yang et al. (2025) "How Chain-of-Thought Works? Tracing Information Flow from Decoding, Projection, and Activation".

---

> ### Author Rebuttal · Authors · 2026-03-31
>
> We thank the reviewer for their detailed comments, which we believe has helped to strengthen the paper.
>
> **W1** We respectfully disagree with the reviewer's characterization of our contribution. We believe we make an important empirical and theoretical contribution -- we provide the first mechanistic account of how verbal confidence is internally computed in LLMs, and ground our findings in a theoretical framework of 2nd order confidence models from neuroscience.
>
> Specifically, our paper: (1) identifies a novel computational phenomenon — the automatic caching of confidence representations during answer generation, before the model has any reason to expect a confidence query; (2) maps the information flow pathway (answer tokens → PANL → CC) through convergent causal evidence; (3) demonstrates that verbal confidence encodes information beyond token log-probabilities, with implications for the first-order vs second-order debate in decision neuroscience (Fleming & Daw, 2017; §1, lines 86–107 and §A, lines 688–714). These findings have direct practical implications for calibration improvement and theoretical implications for understanding metacognitive capacities in LLMs.
>
> **W2** We appreciate this feedback on presentation. We will strengthen transitional text at the start of each experimental section to motivate what question the preceding experiment left open. For example, §2.4 could summarize convergent evidence from steering, patching, and noising, then identify the gap: whether PANL encodes confidence per se or answer content — motivating the swap experiment.
>
> Regarding moving experiments to the appendix: we respectfully argue the full suite is essential in the main text. Each addresses a distinct question — patching (sufficient?), noising (necessary?), swapping (confidence-specific?), probing (what is encoded?), attention blocking (how does information flow?). Convergent evidence across methods is substantially more persuasive than any single experiment.
>
>
> **W3** The reviewer makes an astute point. Our current evidence characterizes what verbal confidence is not rather than what it is. However, establishing that verbal confidence reflects a distinct computation from token log-probabilities is itself important: it means verbal confidence and white-box logprob methods draw on partially independent signals, consistent with second-order models of confidence (Fleming & Daw, 2017; §1, lines 86–107), with direct implications for calibration and confidence elicitation. We have bolstered this analysis with several additional logprob baselines  - please see response to Reviewer #4 (eMyr): W2. A full positive characterization — e.g., how the PANL signal relates to error detection capacity — is a key future direction.
>
> **W4** We have substantially expanded experimental coverage. We now include two additional datasets (BigMath for mathematical reasoning, MMLU for 57 subjects spanning STEM, humanities, and social sciences), and a third model (Mistral Small 3.1 24B, a reasoning model with CoT enabled by default). The core findings replicate across all settings. Please see our responses to Reviewer #1 UYqi (W3; datasets and Q2: CoT) and Reviewer #2 giwM (W2, architecture) for details.
>
> **W5** Scaling factors of 2 and 5 were selected empirically. We have now tested scales of 1 and 10. All four reproduced the same temporal precedence pattern. Scale 1 produced effects comparable to scale 2; scale 10 was ~50% greater than scale 5. We will include plots for all four scales in the revised manuscript.
>
> **W6** We have now tested Mistral Small 3.1 24B (TriviaQA, a reasoning model with CoT enabled by default). The cached retrieval mechanism generalizes to CoT, with PANL retaining its causal role in Magistral. Please see our response to Reviewer #1 UYqi (Q2: CoT question) for details.
>
> **W7** We thank the reviewer for these references. Yang et al. (2024) provides complementary behavioral analysis of verbalized confidence — our work extends this by investigating internal mechanisms. Yang et al. (2025) is relevant to our new CoT findings as they also trace information flow under CoT. We will refer to both in revision.
>
>
> **Key Questions** We appreciate the reviewer's overarching recommendation. Regarding breadth: we have now expanded to three models (Gemma, Qwen, Mistral), three datasets (TriviaQA, BigMath, MMLU), and both CoT and non-CoT conditions (see W4, W6 above). Regarding focus: we believe the two research questions — when confidence is computed and what it represents — are tightly coupled and mutually reinforcing. The "when" question (cached retrieval vs JIT) motivates the causal experiments; the "what" question (beyond logprobs) addresses the nature of what is cached. We will, however, ensure the revised manuscript articulates this connection more explicitly, following the reviewer's suggestion in W2.

---

> > ### Author Rebuttal · Reviewer_ziFg · 2026-04-01
> >
> > You mentioned expanding the models and experiments, but I cannot see the actual experimental results. You could have provided the new data via tables or an anonymous link, and explicitly referenced the corresponding figures or tables in the main text to facilitate a direct comparison.
> >
> > Regarding your response to W2: While this might be the motivation behind your work, it reads as fragmented and disjointed to me. You should provide the specific revised text or, at the very least, a clear revision strategy (of course, please feel free to disregard this if you believe my assessment is incorrect).

---

> > > ### Author Response · Authors · 2026-04-03
> > >
> > > We thank the reviewer for their comments.
> > >
> > > Point 1: A PDF containing Figures that show the results of the additional experiments (see figures 1-5)  can be obtained at:
> > >
> > > https://osf.io/cs7ph/files/2xzvp?view_only=1c6227274bcd49cba75c0147c94697e0
> > >
> > >
> > > Point 2: We appreciate the reviewer's comment and agree that the logical progression of experiments is important to clarify. We propose adding a roadmap figure  that makes the logical progression explicit (see Figure 7 in pdf at link above). We will also strengthen the transition between Sections 2.2 and 2.3, which we agree is the weakest.
> > >
> > > We will include the figures where possible in the main paper (space permitting), otherwise they will be in the Appendix.

---

### Official Review · Reviewer_giwM · 2026-03-09

**Soundness:** 3
**Presentation:** 4
**Significance:** 4
**Originality:** 3
**Overall Recommendation:** 5
**Confidence:** 4

**Summary:**

**Summary**:
This paper investigates how verbalized confidence scores are represented in LMs, focusing on (1) identifying at which point in the generation the confidence scores are formulated and (2) what aspects the confidence score representation encodes. The authors propose two different hypotheses for the first question and conduct *causal experiments* through activation steering and attention blocking to validate the hypothesis. Experiments in a single short-form QA dataset and two models show evidence of confidence representations forming immediately after the answer, as well as immediately before verbalizing their confidence. The results provide compelling evidence that a cached retrieval mechanism is behind the verbalized confidence. Additionally, through statistical analysis of the variance of linear probes on these representations shows that predictiveness of verbal scores is not solely explained by the answer token log probabilities, suggesting that verbalized scores do not merely consider fluency of the answer.


**Soundness**:
The submission is technically sound and all claims are supported. Error bars are provided, which strengthens the relevance of the results. A few additional experiments are included, which improves the generalization of their findings to different model architectures, model size, and prompts with different output formats (linguistic vs numerical, different class cardinality). A few questions about the methodology and generalization to different settings prevail but the validity of the work remains high.

**Presentation**:
The submission is clearly written and well structured. I left a few comments below that could help improve clarity a bit further. I also list a relevant related work which I’d like the authors to comment on.

**Significance**:
The paper addresses an important problem, concerned with a better understanding of how verbalized confidence scores emerge in language models through activation steering.

**Originality**:
This work provides new insights and deepens our understanding of how verbalized confidence emerges.

**Compliance With Llm Reviewing Policy:**

Affirmed.

**Final Justification:**

This paper adopts the mechanistic interpretability tools to shed light on a very relevant question in the NLP domain. While some reviewers raise concerns about the experimental design, I personally thought the experiment design and observed results were sufficient to support the claims. I personally am not super familiar with the literature in mechanistic interpretability so it is possible I have missed something in those experiments.

I understand that some reviewers are concerned about the novelty of this paper and claim it to be an application of the tools in mechanistic interpretability. However, in my view, the paper presents sufficient novelty as it uses the tools to investigate / analyze a previously under explored question -- the mechanism behind LMs providing confidence scores.

My main concerns come from generalizability of the findings to different model sizes and families since the authors include models of different sizes and families, therefore preventing to draw any conclusion when it comes to scaling laws or model's behaviors.

Overall, the paper is well written and organized. I think it is a good paper and recommend acceptance.

**Key Questions For Authors:**

Questions:
1. The experiments are conducted using a two-step approach, wherein the first step requires extracting short answers from the LM, and then collecting the corresponding verbal confidence score with a 2nd prompt. This 2-step procedure contrasts with popular 1-shot verbalized confidence prediction, where verbalized confidence is often requested without seeing the answer first. Can the authors comment on the generalization of their findings to this one step approach ?
2. The findings in the paper are conditioned on inputting the model’s original answer. Section 2.2 performs some experiments to disrupt the model’s ability to identify the correctness of the answer by replacing the original answer token’s activations with mean activations. However, this does not provide insights into how the verbal confidence mechanisms would work if we were to replace the original answers with an answer provided by another model nor about how verbalized confidence works if the incorrect answer is used (which could happen if answers are sampled with temperature!=0). Can the authors comment on this?
3.  In Section 2.2 (lines 172-173) the paper mentions that “effectively preventing the model from assessing whether its answer was correct”. This indicates that there’s an assumption that low confidence and high confidence answers in the calibration set were correlated with correctness/incorrectness. How many low confidence answers were actually incorrect? How do you handle the embeddings of answers with a differing number of tokens? Given that your evidence that the model relies on PANL to determine the confidence score, why not apply the activation patching procedure only to the PANL token?
     - Experiments were replicated using numerical prompt (Section 2.7). However, a well-known limitation of using numerical verbalized confidence estimates is their tendency to concentrate predictions in certain numbers (e.g., 85, 95). Did you find this to affect that “low confidence vs high confidence” calibration set?  Also how does the output of numbers affect the computation of first token - since 1 is prefix of both “1”, “10”, and “100”. How do you control for that?
4. Figure 6  (middle plot) seems to indicate that actually some of the "control' / intermediate tokens have more confidence predictability than PANL for later layers. Which could potentially suggest that there are other layers that may carry important roles for verbalized confidence. In LLMs. Do you have any insight on why that may be? What would happen if we were to run the Activation steering, Activation Patching experiments in the layers where this predictability was detected ?
5. The paper tests 2 prompts: one leading to 10 discrete outputs (Figure 8) and another leading to 101 outputs  (Figure 14). The first prompt is specifically tailored so that each class token has a different starting token, which allows to simplify the analysis to the first observed token. However, a natural question would be how the results would generalize to other semantically similar phrases and whether the observed results are a unique aspect of the prompts used. Or even how the results hold for models with explicit fine-tuning of verbalized scores.
- Section 2.6 runs experiments with a numerical prompt instead of the class prompt, yielding totally different confidence distributions (Figures 9 and Figures 14). While this choice is motivated in the paper, I wonder how generalizable the results of the previous section are to this (new) numeric prompt, and whether this compromises the generalization of the insights for the experiments in Section 2.6. Why haven’t the authors considered an analysis of a few intermediate tokens for the class prompt?
    - By collecting the outputs between 0 and 100, this greatly affects the output space (101 classes as opposed to 10 in the case of linguistic prompts). It is possible that this affects the findings and discrimination ability of the different tokens.

**Suggestions**:
- Would it make sense to include https://aclanthology.org/2025.naacl-long.618/ as a part of the related work since they also adopt a mechanistic approach to investigate whether uncertainty and factuality emerge in the same circuits within LLMs.


**Clarity**: The text is for the most part clear and well-explained. I have a few notes that may further improve the text.
- The expression  “answer token log-probabilities“ (lines 320-321) is ambiguous, since answer can either be the verbalized confidence scores or the original answer. Can the authors clarify what they mean?
- What’s the difference between FCC and CC? In the prompts there only seems to be one colon, so the difference between FCC and CC is unclear.
- The procedure to compute the mean activations from 100 separate calibration trials is not clear (Section C.1.4). My understanding is that the answer tokens do not attend to the confidence trials and, as a result, averaging embeddings over a set of 50 high confidence and 50 low confidence should not change the embedding of the answer tokens at the input. Is the importance of this balanced calibration set the fact that they represent answers for which models had a diverse behavior in terms of calibration?
- How the “confidence change” (lines 182-184) is reported for PANL and CC across layers, since it depends on the confidence output at the end of the prompt.


Typos:
- In a couple of places “focussed” → “focused”

**Limitations:**

I could not find a limitations section although a few limitations could be mentioned.

For instance by highlight the focus of their analysis to the majority answers, since all generations rely on temperature = 0. This may be a form of sampling bias, and may not explain how verbalized confidence scores emerge in a diverse set of cases when other temperature settings are used.

The focusing a subset of prompts, which could limit the sense of how robust these findings are to slight paraphrases. Moreover, numerical verbalized confidence is known to have several issues -- one of them is the saturation of scores in multiples of 5, specifically on 90s or 95. This could affect not only AuROC computation as well as the computation of the logits using only the first token: as the scores "9", "91", "92", ... "99" may all have the exact same prefix "9", as such relying solely on the first tokens logit as a signal may not serve as a measure.

**Strengths And Weaknesses:**

### Strengths:
- S1. Well-structured and comprehensive set of experiments, showing causal relationships between specific tokens and verbal confidence expressions
- S2. Findings are well supported by the conducted experiments and contribute to deepen our understanding of how VC works.
- S3. Experiments that support the generalization of the findings observed in coarse-grained verbalized confidence prompts (using words of estimative probability, 10 classes) to finer-grained numerical prompts (0-100)
- S4. Experiments supporting the generalization of the findings observed to a different architecture and different model scale (Gemma 27B It → Qwen 2.5 7B).


### Weaknesses:
- W1. Experiments are conducted in a single dataset, which may limit the generalizability of the findings. For instance, how would these results generalize to other datasets.
- W2. Experiments to disentangle model scale from architecture are not present: the paper investigates the findings in the context of two models (Gemma3 27B and Qwen 2.5 7B), therefore it cannot properly disentangle or provide insights that may emerge from model scales vs different architecture.
- W3. Please see Questions below for additional considerations

---

> ### Author Rebuttal · Authors · 2026-03-31
>
> We thank the reviewer for their detailed reading of our paper and insightful comments.
>
> **W1** We have now extended our experiments to BigMath (mathematical reasoning) and MMLU (diverse domains including STEM, humanities, and social sciences). Please see response to Reviewer #1 UYqi (W3 point).
>
> **W2: Models** We have now added Mistral Small 3.1 24B (Magistral), a reasoning model with CoT enabled by default, as a third model. This provides a closer scale match with Gemma 3 27B while differing in both architecture and reasoning mode. The core findings — including causal effects at PANL and temporal precedence over CC — replicate (see our response to Reviewer #1 UYqi, Q2). The consistency across 3 architectures (Gemma, Qwen, Mistral) at two scales (7B, 24–27B), and across both CoT and non-CoT settings, strengthens the case that cached retrieval is a general computational strategy.
>
> **Q1** Our two-step procedure is functionally equivalent to one-shot generation: since the Transformer's forward pass is a function of previous tokens, providing the Phase 1 answer as context yields identical residual stream representations at PANL as autoregressive generation (see Figure 1 caption). The two-step design provides experimental control — enabling manipulation of answer token activations while keeping the prompt fixed — without altering the model's computation.
>
> **Q2** It is worth noting, our prompt does not explicitly identify the answer as the model's own. Regarding incorrect answers: ~23% of trials are naturally incorrect, and the model assigns systematically lower confidence to these (Figure 9), suggesting the mechanism operates over answer quality regardless of source. The swap experiment (§2.4) further demonstrates that the confidence signal at PANL is separable from specific answer content, as confidence transfers across unrelated question-answer pairs.
>
> **Q3a** The calibration set (50 high, 50 low confidence trials) creates an uninformative average to disrupt trial-specific information, not to encode correctness.
>
> Each answer token position is replaced with the same mean embedding vector computed across all positions from 100 calibration trials (Eq. 1). Answers with more tokens receive more corrupted positions. This does not affect our conclusions: corruption reduces confidence to floor across all trials regardless of answer length (100% token change rate; Figure 3).
>
> Regarding patching at PANL: this is precisely what we do — corruption is applied to answer tokens, then clean activations are selectively restored at PANL, CC, or PANL+1 (§2.2). The complementary experiment of disrupting PANL alone is our noising experiment (§2.3).
>
> **Q3b** Regarding numeric score clustering: stratified sampling ensures balanced high/low representation despite concentration at high values. For prefix ambiguity, our primary analyses use the categorical prompt where each class has a unique first token. For the numeric prompt, we use the full extracted integer for confidence metrics; first-digit logit difference is used only to confirm the qualitative temporal precedence pattern.
>
> **Q4** The reviewer makes an insightful observation. We address this directly in §2.5 (lines 330–335): information about correctness and verbal confidence is distributed throughout the model — even at positions where causal interventions (steering, patching, noising) have no effect (e.g., PANL+1, FCC -- the colon of the confidence instruction before $CLASS). This dissociation between decodability and causal relevance is well-established in the interpretability literature, which is why we complement probing with causal experiments.
>
> **Q5a** Our results replicate across three prompt formats (categorical, numeric 0–100, minimal 0–9) and three architectures (Gemma, Qwen, Mistral). The unique first-token property is a methodological convenience, not a requirement — numeric prompts lack this property yet replicate the temporal precedence pattern (Figures 16, 19). Testing with paraphrased prompts and fine-tuned models is a valuable future direction.
>
> **Q5b** Complementary attention blocking with the full categorical prompt (§D.1, Figure 18) converges with minimal prompt findings: CC does not compute confidence from scratch, and blocking ALL→last_A produced ~50% change rate, reduced to ~20% when preserving only PANL→last_A. The direct CC→PANL pathway was detectable only in the minimal prompt — the categorical prompt's ~100+ intermediate template tokens provide redundant routing that masks single-edge blocking. We did attempt to trace through these intermediate tokens, but the combinatorial complexity made it intractable to isolate specific pathways. The minimal prompt was designed precisely to reveal this pathway.
>
> **Suggestions, clarify, limitations** We appreciate the reviewer's thoroughness. We will address all remaining clarity suggestions, typos, and limitations (including discussion of temperature=0 and numeric score saturation) in the revised manuscript.

---

> > ### Author Rebuttal · Reviewer_giwM · 2026-04-02
> >
> > Thank you! The rebuttal addresses my concerns. I already thought this was a very well-designed and written paper. The authors addressed both of my concerns namely the (1) additional datasets (BigMath and MMLU), (2) additional model. While the authors do not address the concern directly (Experiments to disentangle model scale from architecture are not present) the addition of a new model does help corroborate the generalization of the findings.

---

> > > ### Author Response · Authors · 2026-04-03
> > >
> > > We are grateful to the reviewer for their comments.
> > >
> > > Should they be of interest, there is a pdf containing figures showing results from the additional model/datasets at:
> > >
> > > https://osf.io/cs7ph/files/2xzvp?view_only=1c6227274bcd49cba75c0147c94697e0
> > >
> > > We will include the figures where possible in the main paper (space permitting), otherwise they will be in the Appendix.

---

### Official Review · Reviewer_UYqi · 2026-03-13

**Soundness:** 2
**Presentation:** 3
**Significance:** 2
**Originality:** 3
**Overall Recommendation:** 4
**Confidence:** 4

**Summary:**

This paper identifies the presence of the explicit confidence signal within Large Language Models and analyzes its passage through the model’s layers. By applying various intervention techniques (activation steering, patching, etc.) to two Language Models (Gemma 3 and Qwen 2.5), the authors show that, indeed, there is a latent confidence signal present at answer token adjacent positions even when the model is not directly prompted to output a verbal confidence score.

**Compliance With Llm Reviewing Policy:**

Affirmed.

**Final Justification:**

I thank the authors for the provided answer. I have also read other reviews and responses to them.

In general, I think that addition of the new material (experiments performed during the rebuttal period) and incorporation of the proposed changes would benefit the article. My main concern with this paper is, as I described in my review, the confusion in the definition of the model's confidence in its answers between two cases: when the model is about to generate an answer and when this answer has already been generated, but the model hasn't yet been asked about its confidence in it (post-hoc, but not post-answer).

This paper explores the second scenario, but the wording in some places implies the first. A clear delimitation between them should be added to the text (potentially in the appendix).

There is another issue with the second scenario -- the fact that the confidence signal is still detected post-hoc (i.e., after the answer is outputted). This may not be the result of some emergent abilities of the model but a remnant of the training procedure. And in this perspective, conclusions about metacognition and introspective awareness are somewhat overextended and, in my opinion, should be made more moderately.

I think that, considering all the proposed improvements, it is possible for me to raise my overall score for this paper to 4.

I apologize for the delayed response.

**Key Questions For Authors:**

1. How would various prompts affect the identified confidence indicators? In particular, what will happen if the model is directly prompted to behave “insecurely/uncertainly” or “confidently/assertively”?
2. Does the behaviour of PANL differ when the model is instructed to perform CoT reasoning?
3. Some minor typos:
- line 317: space between brackets
- make dash length consistent throughout the text (“---“ vs “--“)
- please increase font size on Figure 2

**Limitations:**

Not given. This is a fundamental work in machine learning; potential negative societal impact cannot be directly estimated. Potential limitations of the proposed method include applicability only to LLMs that allow direct access to their internal states and a focus on models without external reasoning.

**Strengths And Weaknesses:**

Strengths:

1. This paper provides an interesting insight into the inner workings of the transformer models. The results of this paper can contribute to the improvement of mechanistic interpretability / circuit analysis of LLMs.
2. This paper includes multiple experiments (patching/noising/activation swapping) to comprehensively study the phenomenon.
3. In general, this paper is well-written, has nice structure, and is easy to follow.

Weaknesses:

1. The results correspond to the general assumption that tokens at the end of the text blocks gather information about preceding passages. In general, the scope of the paper could be expanded.

2. The PLNA token doesn’t fully correspond to the confidence during answer generation because it is outputted after the answer was already generated. This somewhat undermines the main claim about the detected confidence signal not being a post-hoc evaluation. Potentially, a more interesting token to study would be the last token before the answer (the final token of `**Answer:**` in the implemented prompt design) is also a potential key point for answer/confidence processing; however, it got no coverage in the study.

3.   Experiments cover only one dataset. Some different types/formats of questions would be appreciated.

---

> ### Author Rebuttal · Authors · 2026-03-31
>
> We thank the reviewer for their constructive comments, which we believe has enabled us to strengthen the paper.
>
> **W1** While our study is grounded in the observation that end-of-block tokens aggregate information from preceding passages, our contribution does not reduce to it. Specifically, information aggregation at end-of-block tokens is a necessary but not sufficient condition for cached confidence. Granting that PANL aggregates information, three possibilities remain: PANL could aggregate general answer content from which CC computes confidence de novo (JIT); PANL could contain confidence information that is not causally used; or PANL could encode a confidence-specific representation that is causally retrieved at CC. Distinguishing between these requires exactly the causal experiments we report.
>
> Our contribution provides answers to 2 key questions: whether the model automatically computes a confidence-specific representation at PANL — even when it has no reason to expect a confidence query — and whether that representation is causally used during verbalization.
>
> **W2** If we understand correctly, the suggestion is to examine the token immediately preceding the answer (final token of "Answer:"). However, this position cannot encode question-specific confidence because no answer tokens have yet been generated — the model has access to the question but not its own response.
>
> We did examine answer tokens directly (Figures 10 and 16). Steering at the first answer token showed weak effects; steering at the last answer token was effective but weaker than at PANL, and critically confounded — it simultaneously serves as answer content and potential confidence locus, making it impossible to disentangle interventions on confidence from interventions on the answer content from which confidence is derived. PANL provides a clean separation (§2.1, lines 155–164).
>
>
> Regarding the concern that PANL represents a "post-hoc" evaluation: while the last answer token is technically the first position with access to both question and answer, PANL is the first unconfounded position. The key finding is that confidence emerges here automatically, without any prompt to self-evaluate — this is what distinguishes cached retrieval from post-hoc evaluation triggered at CC.
>
> **W3: Regarding Datasets** We agree with the reviewer that broader dataset coverage strengthens the paper. In response to this concern (shared across reviewers), we have extended our experiments to two additional datasets: BigMath, which tests mathematical reasoning, and MMLU, which covers 57 subjects spanning STEM, humanities, social sciences, and other domains. Results on both datasets replicate the core findings  — PANL effects emerging at earlier layers than CC — across steering, patching, and noising experiments in Gemma 3 27B. We will include these results in the revised manuscript.
>
> **Q1** This is an interesting question. Our current experiments vary prompt format (categorical vs. numeric confidence scales) and show that the cached retrieval mechanism is robust to this variation. The reviewer's suggestion — manipulating the model's disposition toward confidence via system-level instructions — is a distinct and complementary question about whether such prompts modulate the same PANL representations we identify, or bypass them entirely. We consider this a valuable direction for future work and will note it in the revised manuscript.
>
> **Q2: Regarding CoT** We have now tested Mistral Small 3.1 24B (Magistral), a reasoning model with CoT enabled by default, on TriviaQA. We replicate the core findings observed in Gemma relating to PANL, including activation patching, activation swap experiments, and linear decoding. Activation noising at PANL did not produce significant disruption — this likely reflects the availability of additional caching sites within the many tokens contained within the reasoning trace, such that ablating a single position is compensated by redundant representations. Consistent with this, our decoding results show a concentration of confidence-predictive information at the final token of the reasoning trace. These results demonstrate that the cached retrieval mechanism generalizes to a third architecture and to a CoT setting, with PANL retaining its causal role alongside distributed trace representations.
>
> **Q3**  We thank the review for noting these, we will correct these in the revised manuscript

---

> > ### Author Rebuttal · Reviewer_UYqi · 2026-04-04
> >
> > I thank authors for the provided response. However, not all of my main concerns were fully addressed, In particular, experiments on other datasets were promised, but their results were not provided (although it may take some time to obtain them).
> >
> > Regarding **W.2**. My concern was indeed about the possibility of confidence signal extraction at the point right before answer is generated (i.e., how confident the model is in what it is about to generate). Right now, the paper and the reply to **W.2** do not provide explain clearly why the detected signal is model's internal confidence, but not the post-hoc evaluation of the generated answer.

---

> > > ### Author Response · Authors · 2026-04-05
> > >
> > > We thank the reviewer for their constructive comments.
> > >
> > >
> > > Point 1: A PDF containing Figures that show the results of the additional experiments (see figures 1-8 and Table 1)  can be obtained at:
> > >
> > >
> > > https://osf.io/n2sjt/files/2cugy?view_only=370e8ed4dd3b4809989882c0b43f4452
> > >
> > >
> > >
> > > Point 2: We thank the reviewer for clarifying this point. The reviewer is asking about the last pre-answer token (the colon following 'Answer', which we term AC) — the position where the model's output distribution determines what answer it is about to generate.
> > >
> > >
> > > We address this at two levels:
> > >
> > >
> > > **1. Log-probability baselines.** The logprob of the first answer token is computed from the logits at the AC position, and thus directly captures the model's pre-answer confidence in the first answer token it is about to generate. In the original paper, we showed that mean answer logprobs (i.e. across all answer tokens) explain little variance in verbal confidence (R²=0.084). Also following a suggestion by Reviewer eMyR, we now include six logprob baselines — including first answer token logprob (R²=0.070, extracted at AC) — and show that PANL activations explain substantial unique variance after controlling for all six combined (R²_unique=0.376; see Table 1 and Figure 6 in supplementary PDF). PANL representations, therefore, go far beyond the pre-answer signal the reviewer enquires about, in predicting verbal confidence.
> > >
> > >
> > > **2. Causal experiments at AC.** We have conducted additional experiments — activation steering, patching, noising, and linear decoding — targeted at the AC position. These show that AC activations do not play a causal role in verbal confidence generation and encode substantially less confidence-related information compared to PANL (see Figure 8 in supplementary PDF). This is expected: the AC position has access only to question tokens, not to the model's own answer, and therefore can only encode question-level properties such as difficulty or domain familiarity — not an evaluation of whether a specific answer is correct.
> > >
> > >
> > > **3. On the nature of PANL as "post-hoc."** The confidence signal at PANL is indeed post-answer — it follows the generated answer and reflects evaluation of it. However, it is not post-hoc in the sense of being reconstructed when the model is later asked to report confidence. The key finding is that confidence is computed at PANL automatically during generation, before the model has any indication that a confidence rating will be requested. This is the distinction our paper draws: between automatic, concurrent evaluation (cached retrieval) and evaluation triggered by the confidence query (just-in-time computation at CC, which our attention blocking experiments rule out).
> > >
> > >
> > > Together, these results reinforce our central finding: verbal confidence reflects an automatic post-answer evaluation at PANL — one that is computed before any confidence query, encodes richer information than pre-answer generation uncertainty at AC, and is subsequently retrieved at CC for verbalization.
> > >
> > >
> > > We will include the figures where possible in the main paper (space permitting), otherwise they will be in the Appendix.

---

### Decision · Program_Chairs · 2026-04-30

**Decision:**

Accept (regular)

**Comment:**

This work presents an interesting and very timely analysis of when and how language models compute verbal confidence. The authors also improved their task coverage in the rebuttal period, which has made the work stronger. I particularly recommend that authors take Reviewer UYqi's comments into account: the scope of the paper could have been broadened by a better task coverage.

I discarded the methodological novelty criticism by Reviewer ziFg in my meta review---applying existing methods to a meaningful and important task is, of course, a solid contribution. I recommend Reviewer ziFg to adjust their expectation for paper contributions in their future reviews.